



# Atmospheric particle abundance and sea salt aerosol observations in the springtime Arctic: a focus on blowing snow and leads

Qianjie Chen[1,10], Jessica A. Mirrielees[1], Sham Thanekar[2], Nicole A. Loeb[3,12], Rachel M. Kirpes[1], Lucia M. Upchurch[4,5], Anna J. Barget[1], Nurun Nahar Lata[6], Angela R. W. Raso[1,7], Stephen M. McNamara[1], Swarup China[6], Patricia K. Quinn[4], Andrew P. Ault[1], Aaron Kennedy[3], Paul B. Shepson[7,8,11], Jose D. Fuentes[2], and Kerri A. Pratt[1,9]

[1] Department of Chemistry, University of Michigan, Ann Arbor, Michigan 48109, United States
[2] Department of Meteorology and Atmospheric Science, Pennsylvania State University, University Park, Pennsylvania 16801, United States
[3] Department of Atmospheric Sciences, University of North Dakota, Grand Forks, North Dakota 58202, United States
[4] Pacific Marine Environmental Laboratory, National Oceanic and Atmospheric Administration, Seattle, Washington 98115, United States
[5] Cooperative Institute for Climate, Ocean, and Ecosystem Studies, University of Washington, Seattle, Washington 98115, United States
[6] Environmental Molecular Sciences Laboratory, Pacific Northwest National Laboratory, Richland, Washington 99352, United States
[7] Department of Chemistry, Purdue University, West Lafayette, Indiana 47907, United States
[8] Department of Earth, Atmospheric, and Planetary Sciences & Purdue Climate Change Research Center, Purdue University, West Lafayette, Indiana 47907, United States
[9] Department of Earth and Environmental Sciences, University of Michigan, Ann Arbor, Michigan 48109, United States
[10] Now at Department of Civil and Environmental Engineering, The Hong Kong Polytechnic University, Hong Kong SAR, China
[11] Now at School of Marine and Atmospheric Sciences, Stony Brook University, Stony Brook, New York 11794, United States
[12] Now at Department of Environment and Geography, University of Manitoba, Winnipeg, Manitoba, Canada

*Correspondence to*: Kerri A. Pratt (prattka@umich.edu)

**Abstract.** Sea salt aerosols play an important role in the radiation budget and atmospheric composition over the Arctic, where the climate is rapidly changing. Previous observational studies have shown Arctic sea ice leads to be an important source of sea salt aerosols, and modeling efforts have also proposed blowing snow sublimation as a source. In this study, size-resolved atmospheric particle number concentrations and chemical composition were measured at the Arctic coastal tundra site of Utqiaġvik, Alaska during spring (April 3 – May 7, 2016). Blowing snow conditions were observed during 25% of the five-week study period and were over-predicted by a commonly used blowing snow parameterization based solely on wind speed and temperature. Throughout the study, open leads were present locally. During periods when blowing snow was observed, significant increases in the number concentrations of 0.01-0.06 μm particles (factor of six, on average) and 0.06-0.3 μm particles (67%, on average), and a significant decrease (82%, on average) in 1-4 μm particles, were observed, compared to low wind speed periods. These size distribution changes were likely caused by the generation of ultrafine



particles from leads and/or blowing snow, with scavenging of supermicron particles by blowing snow. At elevated wind

speeds, both submicron and supermicron sodium and chloride mass concentrations were enhanced, consistent with wind-dependent local sea salt aerosol production. At moderate wind speeds below the threshold for blowing snow, as well as during observed blowing snow, individual sea spray aerosol particles were measured. These individual salt particles were enriched in calcium relative to sodium in seawater, due to the binding of this divalent cation with organic matter in the sea surface microlayer and subsequent enrichment during seawater bubble bursting. The chemical composition of the surface

snowpack also showed contributions from sea spray aerosol deposition. Overall, these results show the contribution of sea spray aerosol production from leads on both aerosols and the surface snowpack. Therefore, if blowing snow sublimation contributed to the observed sea salt aerosol, the snow being sublimed must have been impacted by sea spray aerosol deposition, rather than upward brine migration through the snowpack. Sea spray aerosol production from leads is expected to increase, with thinning and fracturing of sea ice in the rapidly warming Arctic.

**1. Introduction**

The Arctic is warming twice as fast as the rest of the Earth (Cohen et al., 2014). Climate models have identified the important roles of tropospheric aerosols and ozone in the rapid warming of the Arctic (Shindell et al., 2006; Shindell, 2007; Shindell and Faluvegi, 2009; Najafi et al., 2015; Breider et al., 2017). Emissions of sea salt aerosols (SSA) from leads (fractures in the sea ice) over the Arctic have been observed over many decades across all seasons (Radke et al., 1976;

Nilsson et al., 2001; Leck et al., 2002; May et al., 2016; Kirpes et al., 2019). SSA directly scatter solar radiation (Murphy et al., 1998), serve as cloud condensation nuclei (Collins et al., 2013; Quinn et al., 2017; Fossum et al., 2020), and serve as a source of reactive halogens (e.g., Fan and Jacob, 1992; Peterson et al., 2017; McNamara et al., 2019). Recent modeling studies hypothesize that blowing snow could also inject large amounts of SSA into the polar atmospheric boundary layer (Yang et al., 2008; 2019; Huang and Jaeglé, 2017; Huang et al., 2018). While recent Antarctic ground-based observations

have implicated blowing snow as a source of SSA (Giordano et al., 2018; Hara et al., 2018; Frey et al., 2020), in-situ observations are lacking in the Arctic. At Hudson Bay, Canada in March 2008, an eddy covariance particle flux experiment over sea ice reported emissions of 0.3-2 μm particles (aerodynamic diameter) during elevated wind speeds of ~10 m s$^{-1}$ (Whitehead et al., 2012). These particle emissions were attributed to particle resuspension from the sea ice surface, but no chemical measurements were conducted for confirmation (Whitehead et al., 2012). An improved understanding of the

contribution of sea ice leads and blowing snow to the SSA burden in the Arctic will improve prediction of the radiation budget in the changing Arctic climate (Struthers et al., 2011; Browse et al., 2014). It will also provide an assessment of the use of ice core sea salt as a proxy for historical sea ice extent (Levine et al., 2014; Rhodes et al., 2017).

Leads are an important source of Arctic SSA (Scott and Levin, 1972; Radke et al., 1976; Leck and Bigg, 1999; Nilsson et al.,

2001; Leck et al., 2002; Leck and Svensson, 2015; May et al., 2016; Kirpes et al., 2019). Aerosol particles produced from



seawater bubble bursting often contain both inorganic salts and organic species, and are therefore referred to as sea spray aerosols (Quinn et al., 2015). Sea spray aerosol emissions from the open ocean and leads depend strongly on the wind speed magnitude (Nilsson et al., 2001; Monahan and O'Muircheartaigh, 1980). The emissions are also affected by seawater temperature, seawater salinity, ocean depth, surface fetch, surface-active substances, and atmospheric stability (Lewis and

Schwartz, 2004; Grythe et al., 2014). In particular, through experiments using Arctic Ocean water at temperatures from 2 to 5°C, Zábori et al. (2012) found the number concentration of sea spray aerosols > 0.01 µm increased by 22-33% per 1°C of water temperature decrease. Nilsson et al. (2001) reported eddy covariance flux measurements over the Arctic Ocean during summer and found that the sea spray aerosol emission flux over the open leads increased by a factor of five when the 10-m wind speed increased from 2 to 9 m s$^{-1}$. The sea spray aerosol number flux over the open leads was observed to peak at 1-3

µm (optical diameter) and extended into the submicron range (Nilsson et al., 2001). At low wind speeds, lead-based bubble bursting source mechanisms may be important, but are not well understood (Willis et al., 2018). In addition to wind-driven formation, bubbles are suggested to form from gas release during melting of nearby ice (Nilsson et al., 2001), respiration of algae and phytoplankton (Johnson and Wangersky, 1987), and bubble formation below the sea surface driven by sea-air heat exchange (Norris et al., 2011). Recently, May et al. (2016) analyzed three years of particulate inorganic ion concentrations at

Utqiaġvik (Barrow), Alaska. They found year-round enhancements in supermicron sea salt mass concentrations, but negligible change in submicron sea salt mass concentrations, in the presence of nearby leads when the 10-m wind speed exceeded 4 m s$^{-1}$ (May et al., 2016). Through single-particle analysis of aerosol samples collected at 1.5 m above the snowpack near Utqiaġvik when the 10-m wind speeds were greater than 4 m s$^{-1}$, Kirpes et al. (2019) observed sea spray aerosols produced locally from open sea ice leads as the dominant aerosol source in the coastal Alaskan Arctic during winter.

However, the contributions of leads to the Arctic SSA budget are not well-constrained, in part due to the lack of resolution of leads in most numerical models (Wang et al., 2016) and the spatially heterogeneous nature of Arctic sea ice leads (Wernecke and Kaleschke, 2015).

Sublimation of blowing snow could potentially serve as a source of SSA in the Polar Regions (Yang et al., 2008; Huang and

Jaeglé, 2017; Giordano et al., 2018; Frey et al., 2020). In the modeling parameterization proposed by Yang et al. (2018), aerosol particles produced from blowing snow sublimation are assumed to contain only salts and not organics, and are therefore referred to as SSA. During windy conditions, falling or previously fallen snowflakes can bounce and then shatter into smaller snow crystals (Mellor, 1965; Sato et al., 2008; Comola et al., 2017). The physical and chemical properties of the snowpack change in time through metamorphism, compaction, and interactions with the atmosphere (Bartels-Rausch et al.,

2014; Wolff et al., 2022; Domine, 2022). During blowing snow periods, snow grains can be lifted up into the saltation layer (typically <100 mm above the surface; Gromke et al., 2014) through aerodynamic entrainment by turbulent flow (Doorschot and Lehning, 2002) or splash entrainment by collisions of particles (Comola and Lehning, 2017). Then the snow crystals follow parabolic trajectories in the saltation layer, impact with the surface, and produce smaller snow crystals (Doorschot and Lehning, 2002; Comola et al., 2017). The snow crystals in the saltation layer can be transported higher up into the



atmosphere by turbulent eddies (Xiao et al., 2000; Gordon and Taylor, 2009). The blowing snow particle size distribution
($f(d_{BLSN})$), both within and above the saltation layer, has been observed to follow a two-parameter gamma probability density
function ($E1$), with an average diameter ranging from 73 to 262 µm (Antarctica: Frey et al., 2020; Budd, 1966; Nishimura
and Nemoto, 2005; Churchill, Canada: Gordon and Taylor, 2009; French Alps: Nishimura et al., 2014; Wyoming, US:
Schmidt, 1981; 1982). The $f(d_{BLSN})$ is defined as:

$$f(d_{BLSN}) = \frac{e^{-\frac{d_{BLSN}}{\beta}} d_{BLSN}^{\alpha-1}}{\beta^{\alpha-1}\Gamma(\alpha)} \qquad (E1)$$

where $d_{BLSN}$ is the blowing snow particle diameter, $\alpha$ is the shape parameter (from 2-12) that describes the skewness of the
distribution, $\beta$ is the scale parameter (from 20-100 µm), and $\Gamma$ is the gamma function (Frey et al., 2020). The average
blowing snow particle diameter is $\alpha\beta$.

In numerical models, blowing snow is generally initiated when the wind speed is higher than an air temperature-dependent
threshold ($\approx 7$ m s$^{-1}$ at -27 °C), estimated based on observations of drifting and blowing snow over Canadian Prairies (Li and
Pomeroy, 1997). The blowing snow particles are predicted to sublimate to produce water vapor (Déry et al., 1998) and SSA
in the atmosphere (Yang et al., 2008), with a sublimation flux as a function of relative humidity, heat conductivity, and vapor
diffusion (Déry and Yau, 2001). The mass production rate of SSA, assumed to consist solely of salts and no organics, from
blowing snow sublimation is predicted to depend on the snow particle sublimation rate and snow salinity (Yang et al., 2008).
The size distribution of SSA generated from blowing snow sublimation is estimated from the blowing snow particle size,
snow salinity ($\varsigma$), number of SSA produced per blowing snow particle ($N_p$), and percentage of water loss from the blowing
snow particles (Yang et al., 2008; 2019). Assuming complete loss of water from the blowing snow particles, the diameter of
dry SSA particles ($d_{dry}$) from blowing snow sublimation was predicted to follow $E2$ (Yang et al., 2019):

$$d_{dry} = (\frac{1}{N_p})^{\frac{1}{3}} d_{BLSN} (\frac{\varsigma\rho_{ice}}{1000\rho_{NaCl}})^{\frac{1}{3}} \qquad (E2)$$

where $\rho_{ice}$=917 kg m$^{-3}$ and $\rho_{NaCl}$=2160 kg m$^{-3}$ are the density of ice and NaCl, respectively. Previous modeling studies
assumed blowing snow particle size to follow a two-parameter gamma probability density function ($E1$) with an average
diameter of 75-140 µm, snow salinity on the order of 0.01-10 practical salinity units (psu; g salt / kg sea water), 1-10 SSA
particles produced per blowing snow particle, and complete loss of water from the blowing snow particles (Levine et al.,
2014; Rhodes et al., 2017; Huang and Jaeglé, 2017; Huang et al., 2018; Yang et al., 2008; 2010; 2019). However, this does
not consider the chemical complexity of snow composition (Macdonald et al., 2018; Mori et al., 2019; Grannas et al., 2007;
Krnavek et al., 2012), including the presence of organics, dust, nitrate, sulfate, and soot, and requires many assumptions. In
particular, by assuming only sea salt in blowing snow particles, the size and mass of particles produced from blowing snow
sublimation could be underestimated. The wind-blown snow flux is also highly uncertain and difficult to quantify as it is a
complex interaction of wind (speed, direction, gusts, lulls) and snow surface and subsurface conditions (snow grain size,



bonding, hardness, layering) (Sturm and Stuefer, 2013). Thus, large uncertainties exist for modeling SSA production from blowing snow sublimation, primarily due to a lack of observations for model evaluation.

Both blowing snow and leads are proposed to be important sources of SSA in the Arctic, yet observational data are limited to
inform modeling studies. In this study at a coastal Arctic site during spring, leads were omnipresent, and blowing snow was episodic. Blowing snow was identified by the local airport and ceilometer measurements and compared to that predicted by the blowing snow parameterization used in previous atmospheric chemistry modeling studies (e.g. Yang et al., 2008; 2019; Huang and Jaeglé, 2017; Huang et al., 2018). Size-resolved aerosol number concentrations from 0.01 to 5 μm were measured, and uniquely, both bulk (submicron and supermicron) and single-particle composition were measured. In this
work, we refer to "sea salt aerosol" (SSA) when referring to ion chromatography results that only measured inorganic salts, and we refer to "sea spray aerosol" when referring to microscopy results that showed individual particles with composition consistent with those formed from seawater bubble bursting (Ault et al., 2013). We compare ambient particle abundance and chemical composition observed for different wind speed conditions and discuss the influences of episodic blowing snow and omnipresent leads.

## 2. Methods

### 2.1 Site description and meteorological conditions

This study covers the period between April 3 and May 7, 2016, during the Photochemical Halogen and Ozone Experiment: Mass Exchange in the Lower Troposphere (PHOXMELT), which was conducted at a coastal tundra site near Utqiaġvik, Alaska (71.275°N, 156.641°W) about 5 km from the Arctic Ocean. The sampling site was snow-covered throughout the
whole study period. Figure 1 shows an example satellite image of the North Slope of Alaska (including the location of Utqiaġvik), as well as the bordering Chukchi and Beaufort Seas, on April 6, 2016. This period was characterized as a clean period not affected by local anthropogenic pollution based on concurrent nitrogen oxides ($NO_x$) and wind direction measurements (McNamara et al., 2019). The wind direction was generally between 0° and 100°, indicating air masses coming from the Beaufort Sea (Fig. 1).




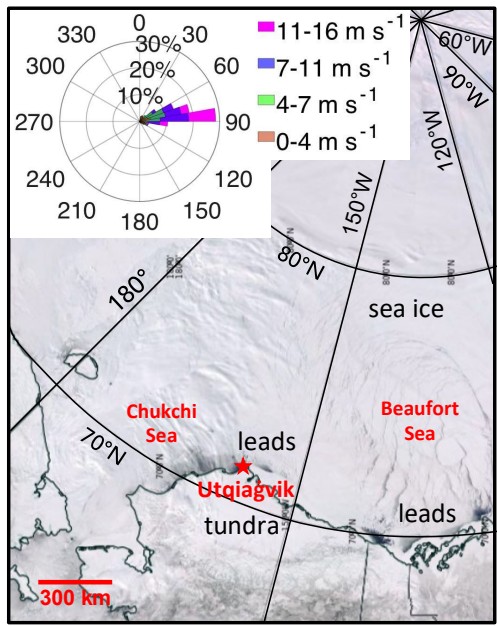

**Figure 1.** Image of the North Slope of Alaska and Beaufort and Chukchi Seas on April 6, 2016 from the Terra satellite Moderate Resolution Imaging Spectroradiometer (MODIS). The dark regions and cracks correspond to leads surrounded by sea ice. The inset plot depicts the percentage distribution of observed wind speeds and directions, measured at a 9.7 m height at the Utqiaġvik tundra site from April 3 to May 7, 2016. This figure is modified from NASA Worldview (https://worldview.earthdata.nasa.gov/).

Two three-dimensional sonic anemometers (model CSAT3, Campbell Scientific Inc., Logan, UT) were installed at 1.3 m and 9.7 m above the ground, respectively, to measure the air temperature ($T$) and wind speed components ($u$ = zonal wind speed, $v$ = meridional wind speed, and $w$ = vertical wind speed) at 10 Hz. The 30-min average friction velocity ($u* = \sqrt[4]{\overline{u'w'^2} + \overline{v'w'^2}}$) was calculated from the turbulence data, where $u'$, $v'$, and $w'$ are deviations from the mean wind speed. Relative humidity with respect to liquid water ($RH_{aq}$) was obtained from the RH sensors at 1.5 m and 10.4 m above the ground, respectively. We converted $RH_{aq}$ to relative humidity with respect to ice ($RH_{ice}$) via $E3$, following the Clausius-Clapeyron equation (Seinfeld and Pandis, 2016):

$$RH_{ice} = RH_{aq} \frac{e^{\frac{L_v}{R_v}(\frac{1}{T_0} - \frac{1}{T})}}{e^{\frac{L_s}{R_v}(\frac{1}{T_0} - \frac{1}{T})}} \qquad (E3)$$





where $L_v$ (= $2.50\times10^6$ J kg$^{-1}$ at 0°C) and $L_s$ (= $2.85\times10^6$ J kg$^{-1}$) are the latent heat of vaporization and sublimation, respectively; $R_v$ (= 461.5 J kg$^{-1}$ K$^{-1}$) is the gas constant for water vapor; $T_0$ = 273.15 K. The threshold wind speed ($U_t$; m s$^{-1}$) for blowing snow to occur was calculated in $E4$, as defined by Li and Pomeroy (1997):

$$U_t = 6.975 + 0.0033(T + 27.27)^2 \qquad (E4)$$

where $T$ (°C) is the air temperature measured at 1.3 m, to be consistent with Li and Pomeroy (1997) who used air temperature at 2 m. The standard error of $U_t$ reported by Li and Pomeroy (1997) was 1.97 m s$^{-1}$.

We define blowing snow (BLSN) wind periods as when the wind speed recorded at 9.7 m above the surface ($U_{9.7m}$) exceeded $U_t$ (i.e., $U_{9.7m} \geq U_t$) (Table 1), to be consistent with Li and Pomeroy (1997) who used 10-m wind speed ($U_{10m}$) $\geq U_t$. The non-

blowing snow (NBLSN) wind periods include two sub-categories: moderate wind periods: 4 m s$^{-1}$ $\leq U_{9.7m} < U_t$; and low wind periods: $U_{9.7m} < 4$ m s$^{-1}$ (Table 1). The wind speed of 4 m s$^{-1}$ is generally considered as the onset wind speed for sea spray aerosol production over the open ocean (Monahan and O'Muircheartaigh, 1986). As such, May et al. (2016) previously used 4 m s$^{-1}$ as the threshold wind speed to separate moderate wind and low wind categories when analyzing SSA abundances at Utqiaġvik, AK.


**Table 1.** Definitions of observed blowing snow (BLSN), falsely predicted BLSN, moderate wind speed, and low wind speed periods. The BLSN wind and non-blowing snow (NBLSN) wind periods are separated based on the comparison of measured 9.7 m wind speed ($U_{9.7m}$) with a calculated threshold blowing snow wind speed ($U_t$) (Li and Pomeroy, 1997). Time covered is the percentage of time classified in each period during the study.

| Category | | Definition | Time covered |
|---|---|---|---|
| BLSN wind | Observed BLSN | $U_{9.7m} \geq U_t$; blowing snow observed by the airport or ceilometer | 25.3% |
| | Falsely predicted BLSN | $U_{9.7m} \geq U_t$; blowing snow not observed by the airport or ceilometer | 19.2% |
| NBLSN wind | Moderate wind | 4 m s$^{-1}$ $\leq U_{9.7m} < U_t$ | 29.2% |
| | Low wind | $U_{9.7m} < 4$ m s$^{-1}$ | 26.3% |


Blowing snow observations occurred at two nearby stations during the study period. Hourly blowing snow events were reported from the Wiley Post-Will Rogers Memorial Airport (71.285°N, 156.769°W) (automated monitoring station; https://www.wunderground.com/; last access: 10/03/2019) located at ~5 km northwest of our field study site. These observations follow the criteria that snow is lofted by the wind to a height of > 1.8 m and horizontal visibility is < 11 km

(Federal Meteorological Handbook, 2019). Note that the airport did not report blowing snow events when it was snowing



simultaneously. Blowing snow events were also identified from ceilometer measurements at the Department of Energy Atmospheric Radiation Measurement (DOE ARM) Observatory (71.323°N, 156.608°W) located 5.8 km north of our sampling site across flat tundra. These events were identified by meeting both of the following criteria (Gossart et al., 2017; Loeb and Kennedy, 2021): (1) backscatter in the lowest usable bin (about 10-20 m above the ground) was larger than clear

sky threshold and decreased with height; (2) $U_{10m} \geq U_t$ and 2 m visibility < 7.5 km. Note that the ceilometer observations only identified deep blowing snow events (>10-20 m above the ground). Within the BLSN wind periods, we define "observed BLSN" periods as when blowing snow was observed by either the airport or the ceilometer; "falsely predicted BLSN" periods correspond to times when BLSN was not observed but was predicted based on $U_{9.7m} \geq U_t$ (Table 1).

**2.2 Aerosol size distribution and chemical composition**

Size-resolved particle number concentrations from 0.010 to 0.420 μm (electrical mobility diameter, $d_m$, 13 size bins) were measured using a scanning mobility particle sizer spectrometer (SMPS; NanoScan SMPS Model 3910, TSI Inc., Shoreview, MN) from April 4 to May 7 (Fig. S1). The SMPS sampled air at the flow rate of 0.75 L min$^{-1}$ through 3 m long, 0.64 cm inner diameter copper tubing that was placed at 3 m above the ground. For the SMPS particle diameter range, the sampling line loss was estimated to range from 4% for 0.420 μm particles to 25% for 0.010 μm particles (Willeke and Baron, 1992).

An optical particle sizer spectrometer (OPS; Model 3330, TSI Inc., Shoreview, MN) measured the size range of 0.3-10 μm (optical diameter, $d_o$, 16 size bins) from April 18 to May 7 (Fig. S1). The OPS sampled air at the flow rate of 1.0 L min$^{-1}$ through 3.5 m long, 0.64 cm inner diameter copper tubing that was placed at 3 m above the ground. For the OPS particle diameter range, the sampling line loss was estimated to increase from 1% for 0.3 μm particles, 25% for 5.3 μm particles, to 63% for 10 μm particles (Willeke and Baron, 1992). Particles with diameters larger than 5.3 μm were not used for this data

analysis because of the lower concentration and limited statistics. Size-dependent sampling line losses were not corrected in the data presented, which focuses on relative changes at a given size during different time periods. Short increases in total SMPS particle concentrations greater than 500 particles cm$^{-3}$ lasting for fewer than 20 minutes were likely associated with local combustion emissions and were therefore removed from the data analysis. In particular, spikes in total SMPS particle concentrations (> 2000 particles cm$^{-3}$) were observed frequently on April 24 due to nearby generator use, and therefore, this

day was also removed from the data analysis (Fig. S1). The average total SMPS particle concentration, with these spikes removed, during the focus period was 200±80 particles cm$^{-3}$. At the National Oceanic and Atmospheric Administration (NOAA) Barrow Observatory (71.323°N, 156.611°W), located adjacent to the DOE ARM Observatory, total particle (> 0.014 μm) number concentrations were measured by a condensation particle counter (CPC) at 11 m above the ground during our study period (https://www.esrl.noaa.gov/gmd/obop/brw/). The CPC total particle number concentrations were ~10%

higher than that from SMPS, due to the wider size range, and showed a correlation coefficient of $R^2$=0.88 (CPC total particle number concentrations *vs.* SMPS total particle number concentrations), showing agreement in temporal variability (Figs. S1 and S2).





For bulk inorganic ion analysis, submicron (aerodynamic diameter, $d_a$ <1.1 μm) and supermicron ($d_a$ from 1.1-10 μm)
particles were collected on Millipore Fluoropore filters (pore size of 1.0 μm) using a Berner-type multijet cascade impactor
sampling at 30 L min$^{-1}$ from 10 m above the ground at the NOAA Barrow Observatory (Quinn et al., 2002). Note that the
cascade impactor size cut is 1.1 μm, and thus we define submicron particles as $d_a$ <1.1 μm and supermicron as $d_a$ = 1.1-10
μm in this study. Daily submicron particle samples were collected, while supermicron particles were collected over 6-7 days
each during the study period. To extract particles from the filters, the filters were wetted with 1 mL of spectral grade
methanol, followed by adding 5 mL of distilled deionized water to the solution and sonicating for 30 min. The extracted
particles were analyzed for bulk inorganic ion (Cl$^-$, Br$^-$, NO$_3^-$, SO$_4^{2-}$, Na$^+$, NH$_4^+$, K$^+$, Mg$^{2+}$, Ca$^{2+}$) concentrations using ion
chromatography, following the method of Quinn et al. (2002). The limits of detection (LODs) were 0.0002 μg m$^{-3}$ for
submicron and supermicron Na$^+$ and Cl$^-$ concentrations and 0.0001 μg m$^{-3}$ for submicron and supermicron Br$^-$, NO$_3^-$, SO$_4^{2-}$,
NH$_4^+$, K$^+$, Mg$^{2+}$, and Ca$^{2+}$ concentrations. In total, 33 submicron samples (April 3 – May 7, 2016) and 4 supermicron
samples (April 7 – May 4, 2016) were collected and analyzed for this study.

For two observed BLSN periods (April 6 07:00–19:00 Alaska Daylight Time (AKDT) and April 6 19:00 – April 7 07:00)
and two moderate wind periods (April 15 19:00 – April 16 07:00 and April 16 08:00–20:00), particles were collected using a
microanalysis particle sampler (MPS, California Measurements) sampling at 2 L min$^{-1}$ (including 1 L min$^{-1}$ of HEPA-filtered
dilution flow) with 0.48 cm inner diameter copper tubing at 3 m above the ground. The MPS was loaded with transmission
electron microscopy (TEM) grids (copper Type-B Formvar grids, Ted Pella Inc) to collect particles on stage 3 ($d_a$ 0.07–0.40
μm) and with aluminum foil on stage 2 ($d_a$ 0.40–2.8 μm). However, insufficient particle loading on stage 2 inhibited
computer-controlled scanning electron microscopy with energy dispersive X-ray spectroscopy (CCSEM-EDX) analysis of
the April 6 19:00 – April 7 07:00 sample. In total, 7,094 individual particles (2,963 particles on stage 3 and 4,131 particles
on stage 2) were analyzed by CCSEM-EDX using a FEI Quanta environmental SEM with a field emission gun operating at
20 keV and high angle annular dark field transmission detector. The X-ray spectra detected by an EDX spectrometer
(EDAX, Inc) provided the relative abundance of elements C, N, O, Na, Mg, Al, Si, P, S, Cl, K, Ca, Ti, Fe, Ni, and Zn present
in the individual particles. Note that C and O cannot be quantified in particles on the TEM grids (stage 3), and Al cannot be
quantified in particles on the aluminum foil (stage 2). The K-means clustering of the individual stage 3 particle EDX spectra
(Ault et al., 2012) resulted in 50 clusters, which were grouped into seven particle types, including fresh sea spray aerosol,
partially aged sea spray aerosol, fly ash, organic+sulfate aerosols, mineral dust, soot, and potassium+sulfur, based on
elemental composition, morphology, and comparison of cluster EDX spectra with particle classes in previous EDX studies
(Kirpes et al., 2018; Bondy et al., 2018). The K-means clustering of the individual stage 2 particle EDX spectra resulted in
10 clusters, which were grouped into three particle types: fresh sea spray aerosol, potassium chloride, and mineral dust.
Elements with higher atomic number (Z) than Al (e.g., Na, Cl) appear brighter on the Al tape (stage 2), while the dominant
elements in organic particles (C, O) are lower Z than Al and appear darker. The CCSEM analysis can only focus on either
brighter particles than the Al background or darker particles, and given the focus of the manuscript, we analyzed focusing on





the brighter (sea salt aerosol) particles with CCSEM. As a result, the particle percentages reported for stage 2 represent upper limits. The projected area diameter ($d_{pa}$) obtained from CCSEM-EDX spectra is often larger than the aerodynamic diameter

due to particle deformation and spreading upon impaction, and also depends on particle viscosity (Hinds, 2012; Sobanska et al., 2014; Olson et al., 2019).

**2.3 Tundra surface snowpack**

Twenty-one surface snow (top 2 cm) samples were collected at the tundra field study site on a daily basis between April 5 and May 7, 2016 using a polypropylene scoop while wearing disposable polyethylene gloves. Snow samples were stored in a

-40°C freezer both in Utqiaġvik and at the University of Michigan and were transported frozen. Snow samples were melted immediately prior to ion chromatography measurements of cations (Dionex ICS-1100; $Na^+$, $K^+$, $Mg^{2+}$, and $Ca^{2+}$) and anions (Dionex ICS-2100; $Cl^-$, $SO_4^{2-}$, $Br^-$, $NO_3^-$). Deterioration of snow samples for ion concentration measurements over the freezer storage period is expected to be negligible (Li, 1993). The 3σ LODs for the measured ions are: $Cl^-$ (0.03 μM), $SO_4^{2-}$ (0.06 μM), $Br^-$ (0.01 μM), $NO_3^-$ (0.005 μM), $Na^+$ (0.07 μM), $K^+$ (0.08 μM), $Mg^{2+}$ (0.03 μM), and $Ca^{2+}$ (0.13 μM).

**3. Results and discussion**

**3.1. Meteorological conditions and blowing snow occurrence**

Modeling studies calculate a BLSN wind speed threshold to simulate the occurrence of blowing snow events (Yang et al., 2008; 2019; Huang and Jaeglé, 2017; Huang et al., 2018). However, few observational studies have evaluated the accuracy of this method (Gossart et al., 2017; Frey et al., 2020). Therefore, in this study, we compare the observation of BLSN, using

two methods, with this calculation, for the period of April 3 to May 7, 2016 at the coastal field study site near Utqiaġvik, Alaska. As described in Section 2, BLSN wind periods were calculated to occur when the measured 9.7 m wind speed ($U_{9.7m}$) exceeded the air temperature-dependent threshold value ($U_t \approx 7$ m s$^{-1}$ at 1.3 m air temperature ($T_{1.3m}$) of -27°C) (Li and Pomeroy, 1997).

Just over half (57%) of the calculated BLSN wind periods were observed at the airport or by the nearby ceilometer to experience blowing snow, suggesting the occurrence of blowing snow is not always a function of only wind speed and air temperature, and may depend on other factors such as snow size, shape, and density and wind gusts, as previously observed in Antarctica (Gossart et al., 2017) and the Canadian Prairies (Li and Pomeroy, 1997). Among all calculated BLSN wind periods, 47% of these times were identified as observed blowing snow events by the airport, and 26% were identified by

ceilometer measurements. It should be noted that the airport automated station did not report blowing snow events when it was snowing simultaneously, but this covered only 5% of the time during calculated BLSN wind periods. In addition, the ceilometer only identified deep blowing snow events (>10-20 m above the ground), such that these observations could miss shallow blowing snow events. Importantly, blowing snow events were rarely observed by the airport (1% of the study





period) and ceilometer (2% of the study period) during calculated non-blowing snow (NBLSN) wind periods (when $U_{9.7m} <$

$U_t$). Based on these results, using wind speed and temperature as the only criteria for the onset of blowing snow overestimates the occurrence of blowing snow at our coastal Arctic site and suggests likely overestimates in Arctic atmospheric chemistry modeling studies using this parameterization.

Falling and surface snow properties (e.g. size, shape, and density), snow age (metamorphism and dry deposition impacts),

and snow surface roughness also affect blowing snow occurrence (Li and Pomeroy, 1997) and likely impacted the occurrence of blowing snow during this study, leading to the parameterization overestimation. Snowfall events were recorded by the airport during the study period (April 3 - May 7, 2016), as shown in Figure 2b. The most recent snowfall event before the study period occurred on March 26. After that, there was no snowfall at the site until April 10. Note that 61% of the observed BLSN periods were from April 3-10, indicating aged (>8 days) tundra snowpack during the observed

BLSN periods. For the rest of the observed BLSN periods (39%), the tundra snowpack age was < 2 days (Fig. 2b). In comparison, to consider the impacts of snow age on blowing snow sublimation, atmospheric chemistry models generally have applied a scaling factor when calculating a blowing snow sublimation flux and assumed a snow age of 1.5 - 5 days (e.g. Yang et al., 2008; Levine et al., 2014; Huang and Jaeglé, 2017), representing an uncertainty in these models.

The time period of focus herein is divided into 4 categories (Table 1), including: 1) observed BLSN ($U_{9.7m} \geq U_t$; blowing snow observed by the airport or ceilometer), 2) falsely predicted BLSN ($U_{9.7m} \geq U_t$; blowing snow not observed by the airport or ceilometer), 3) moderate wind (4 m s$^{-1}$ $\leq U_{9.7m} < U_t$), and 4) low wind ($U_{9.7m} < 4$ m s$^{-1}$) periods. Sea spray aerosol emissions occur at moderate wind speeds and above, coinciding with BLSN wind speeds. Nilsson et al. (2001) previously observed that the sea spray aerosol emission flux in the summertime Arctic increased by a factor of 157 when the 10-m wind

speed increased from 3 to 14 m s$^{-1}$. During the study herein, the observed BLSN, falsely predicted BLSN, moderate wind, and low wind periods covered 25.3%, 19.2%, 29.2%, and 26.3%, respectively, of the time (Figure 2).



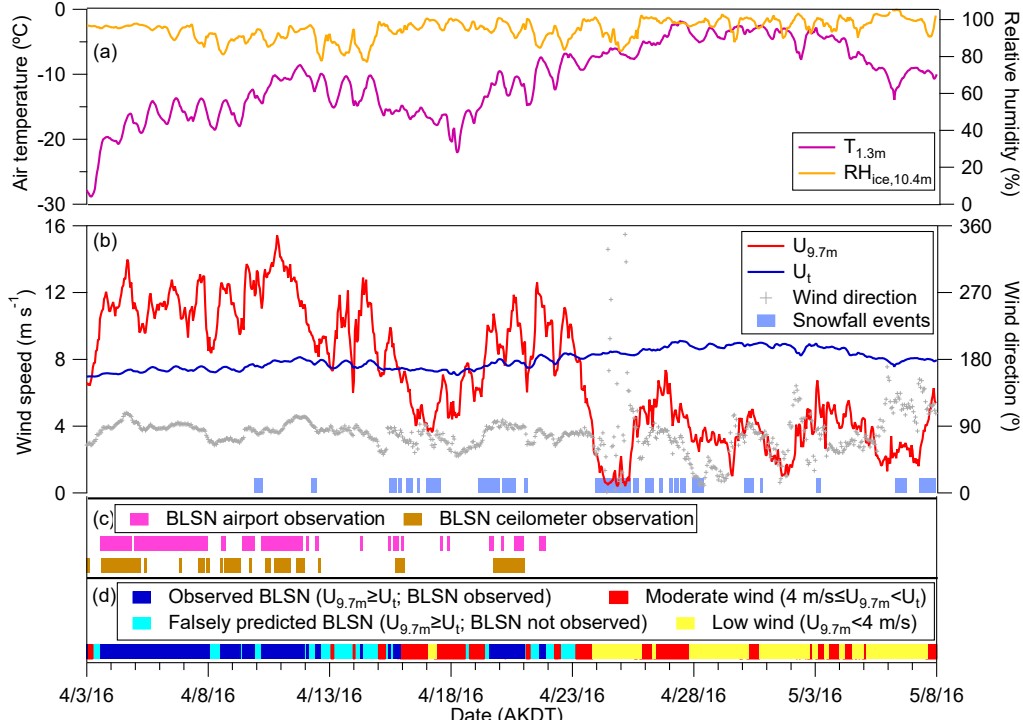

**Figure 2**. Time series of (a) air temperature at 1.3 m ($T_{1.3m}$), relative humidity with respect to ice at 10.4 m ($RH_{ice,10.4m}$), (b)
wind speed ($U_{9.7m}$) and wind direction at 9.7 m, calculated blowing snow threshold wind speed ($U_t$), and snowfall events
recorded by the Utqiaġvik, Alaska airport, (c) observations of BLSN at the airport and from the DOE ARM Observatory
ceilometer measurements, and (d) observed BLSN ($U_{9.7m} \geq U_t$ and BLSN observed by either the airport or ceilometer),
falsely predicted BLSN ($U_{9.7m} \geq U_t$ and BLSN not observed by the airport or ceilometer), moderate wind speed (4 m s$^{-1}$ $\leq$
$U_{9.7m} < U_t$), and low wind speed ($U_{9.7m} < 4$ m s$^{-1}$) periods.

The meteorological conditions, including air temperature, wind speed, relative humidity, and friction velocity, for the
observed BLSN, falsely predicted BLSN, moderate wind, and low wind periods, were compared (Table 2). During observed
BLSN periods, average $U_{9.7m}$ was 11±2 (1σ) m s$^{-1}$, and average $T_{1.3m}$ was -14±3 (1σ) °C; this corresponds to a slightly higher
wind speed ($p < 0.05$), but slightly lower temperature ($p < 0.05$), than for the falsely predicted BLSN periods (10±1 m s$^{-1}$, -
13±4 °C) (Table 2). In comparison, average $U_{9.7m}$ and $T_{1.3m}$ were 6±1 m s$^{-1}$ and -10±6 °C, respectively, during moderate wind
periods and 3±1 m s$^{-1}$ and -7±3 °C during low wind periods. The friction velocity, $u^*$, a measure of atmospheric shear stress,
is an important variable in explaining the occurrence of blowing snow. The saltation process and subsequent lifting of snow
particles are initiated by turbulent wind shear stress exerted on the snowpack rather than the magnitude of the wind (Mann et





al., 2000). Based on snow drift density observations on Antarctic sea ice during winter, Frey et al. (2020) obtained an

average threshold blowing snow friction velocity ($u_t^*$) of 0.37 (ranging from 0.08 to 0.58) m s⁻¹. In this study, the average $u^*$ measured at 1.3 m ($u_{1.3m}^*$) was 0.41±0.06 (1σ) m s⁻¹ during observed BLSN periods (Table 2), consistent with Frey et al. (2020). The relative humidity with respect to ice at 10.4 m ($RH_{ice,10.4m}$ = 93±4%) was below saturation during observed BLSN periods, supporting sublimation of blowing snow upon lofting. The average $u_{1.3m}^*$ was 17% lower during falsely predicted BLSN periods (0.34±0.06 m s⁻¹), compared to observed BLSN periods. The average $RH_{ice,10.4m}$ was 90±6% during

falsely predicted BLSN, similar to the observed BLSN periods. Therefore, while elevated $u^*$ was observed during calculated BLSN periods (Table 2), the occurrence of blowing snow was not guaranteed, perhaps due to physical characteristics of the snowpack. However, it is also possible that blowing snow occasionally occurred during these "falsely" predicted BLSN periods but was not observed by the airport (i.e., overlapping with snowfall) or ceilometer (i.e. episodes of drifting snow < 10 m deep).


**Table 2.** Meteorological conditions during observed BLSN, falsely predicted BLSN, moderate wind speed, and low wind speed periods. $T$, $U$, and $u^*$ (average ± 1σ) are air temperature, wind speed, and friction velocity, at 1.3 m and 9.7 m above ground, respectively. $RH_{ice}$ is the relative humidity with respect to ice at 1.5 m and 10.4 m.

| Category | Time covered | $T_{1.3m}$ (°C) | $T_{9.7m}$ (°C) | $RH_{ice,1.5m}$ (%) | $RH_{ice,10.4m}$ (%) | $U_{1.3m}$ (m s⁻¹) | $U_{9.7m}$ (m s⁻¹) | $u_{1.3m}^*$ (m s⁻¹) | $u_{9.7m}^*$ (m s⁻¹) |
|---|---|---|---|---|---|---|---|---|---|
| Observed BLSN | 25.3% | -14±3 | -15±3 | 95±3 | 93±4 | 9±1 | 11±2 | 0.41±0.06 | 0.44±0.07 |
| Falsely predicted BLSN | 19.2% | -13±4 | -13±4 | 92±4 | 90±6 | 8±1 | 10±1 | 0.34±0.06 | 0.34±0.07 |
| Moderate wind | 29.2% | -10±6 | -10±6 | 98±2 | 97±3 | 4±1 | 6±1 | 0.18±0.04 | 0.19±0.05 |
| Low wind | 26.3% | -7±3 | -7±4 | 98±5 | 98±5 | 2±1 | 3±1 | 0.10±0.04 | 0.12±0.04 |

**3.2. Particle size distributions**

To investigate the impacts of blowing snow and leads on ambient aerosol particles of different sizes, we compare size-resolved particle number concentrations for the different wind speed categories described above. The average size distributions of ambient particle number concentrations measured by the SMPS and OPS near Utqiaġvik are shown in Figure 3. The particle number concentration peaked at 0.12 μm $d_m$, consistent with previous observations at Utqiaġvik and other

Arctic sites during spring (peaking at 0.1-0.2 μm $d_m$; Freud et al., 2017; Croft et al., 2016). For each size bin from 0.01 to 0.06 μm $d_m$ (6 size bins), the average particle number concentrations during observed BLSN periods were 2-13 times (6 times, on average) higher than low wind periods (all significant at 95% confidence levels; $p < 0.05$), 2-12 times (5 times, on average) higher than moderate wind periods ($p < 0.05$), and 37-126% higher than falsely predicted BLSN periods ($p < 0.05$).





In comparison, for each size bin from 0.06 to 0.3 µm $d_m$ (6 size bins), the average particle number concentrations during
observed BLSN periods ranged from 45 to 109% (67%, on average) higher than low wind periods ($p < 0.05$), and from 17 to
80% (31%, on average) higher than moderate wind periods ($p < 0.05$). Compared to falsely predicted BLSN periods, the
average particle number concentrations during observed BLSN periods were between 6 and 16% (10%, on average) higher
within 0.06-0.1 µm $d_m$ (3 size bins) ($p < 0.05$), but were not significantly different from 0.1-0.3 µm $d_m$ (3 size bins) ($p >$
0.05). Therefore, overall, the number concentrations of 0.01-0.3 µm particles were higher at elevated wind speeds.


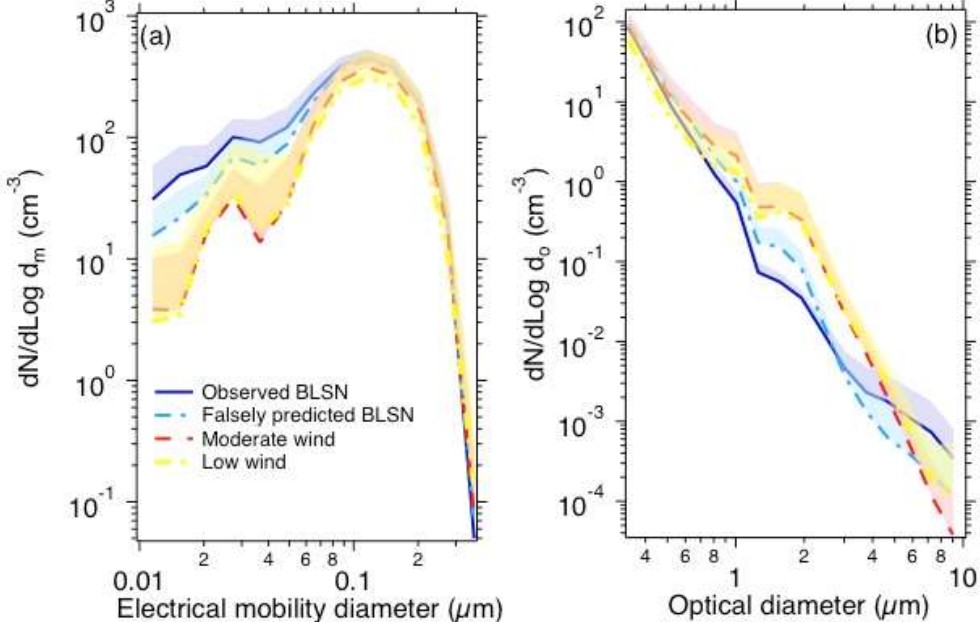

**Figure 3.** Size distributions of particle number concentrations during observed BLSN, falsely predicted BLSN, moderate
wind, and low wind periods from (a) SMPS measurements (0.01-0.42 µm electrical mobility diameter; April 4 - May 7) and
(b) OPS measurements (0.3-10 µm optical diameter; April 18 - May 7). The shadings are the upper bounds of the standard
deviation of dN/dlog$d_m$ (or dN/dlog$d_o$). Since the y axis is on a log scale, only the upper bound of the standard deviation is
shown. Particles larger than 5.3 µm experienced more than 25% sampling line losses and thus were not used in the discussed
data analysis because of the impact of reduced statistics.

Both leads and blowing snow could potentially contribute to the increase in particle number concentrations from 0.01 to 0.3
µm $d_m$ during observed BLSN periods. Yang et al. (2008; 2019) proposed that the size and number of SSA produced from
blowing snow sublimation depend on the size and salinity of blowing snow particle, as well as the percentage of water loss


per blowing snow particle. If we approximate the salinity of the local tundra surface snowpack based on measured [Na$^+$] and [Cl$^-$] (~0.010 g kg$^{-1}$) and assume that only one SSA particle is produced from one blowing snow particle (following Yang et al. (2008)), blowing snow particles would need to have diameters of ~0.6-19 µm to produce SSA particles with diameters of

0.01-0.3 µm. Using the approximate salinity of the surface snowpack on thin first-year sea ice during springtime over the Beaufort Sea (~3.5 g kg$^{-1}$; Krnavek et al., 2012), the same calculation yields 0.09-3 µm blowing snow particles sublimating into 0.01-0.3 µm SSA particles. It is unknown whether blowing snow particles less than 3 µm exist in the atmosphere, as turbulence efficiently removes small crystals in the saltation layer that would otherwise impact with the surface to produce smaller fragments (Comola et al., 2017). Previous field studies showed blowing snow particles larger than ~20 µm (lower

diameter method detection limit) generally following two-parameter gamma probability density functions (*E*1) with average diameters ranging from 73 to 262 µm (Frey et al., 2020; Budd, 1966; Schmidt, 1981; 1982; Nishimura and Nemoto, 2005; Gordon and Taylor, 2009; Nishimura et al., 2014). Modeling studies adapted these empirical functions by using an average blowing snow particle diameter from 75 to 140 µm and extrapolating the blowing snow particle size to < 20 µm, but the smallest blowing snow particle diameter considered is not stated (Yang et al., 2008; 2019; Huang and Jaeglé, 2017; Huang et

al., 2018).

Some atmospheric chemistry modeling studies have assumed that multiple SSA particles are produced from the sublimation of one larger blowing snow particle (Huang and Jaeglé, 2017; Huang et al., 2018; Yang et al., 2019). If blowing snow contributed to the elevated 0.01 to 0.3 µm particles observed in this study, this could provide an explanation if particles

present on and within a larger (> 20 µm) blowing snow particle do not coagulate during sublimation. Previous atmospheric chemistry modeling studies have assumed blowing snow particles contain only sea salt, which is not consistent with the fact that the Arctic snowpack contains a variety of chemical components, including organics, dust, nitrate, sulfate, and soot (e.g., Macdonald et al., 2018; Mori et al., 2019; Grannas et al., 2007; Krnavek et al., 2012). Previous measurements of the size distribution of particles present within Utqiaġvik snowmelt showed the dominance of < 0.3 µm particles (Nazarenko et al.,

2017), similar to the atmospheric particle size distribution. Consistent with this complexity, low temperature microscopy studies have shown that many particles can reside within the snow grain, at the surface, and at grain boundaries (Domine et al., 2008; Bartels-Rausch et al., 2014; Magono et al., 1979).

During the study period, the prevailing northeasterly and easterly winds were from the direction of the Beaufort Sea, where

open leads were omnipresent (Fig. 1). Sea spray aerosols as small as 0.01 µm have been observed to be generated from coastal breaking waves at wind speeds of ~ 7 m s$^{-1}$, with 60%, by number, of the 0.01-8 µm sea spray aerosols smaller than 0.1 µm (Clarke et al., 2006). Nilsson et al. (2001) presented eddy-covariance flux measurements over the Arctic Ocean during summer and found the emission flux of sea spray aerosols > 0.01 µm ($F_{SSA}$) from open leads increased by 5 times when the wind speed ($U_{10m}$) increased from 2 to 9 m s$^{-1}$. In this study, the field site was ~7 km downwind of the nearest lead,

and this corresponds to an ~10 min travel time for sea spray aerosols during observed and falsely predicted BLSN periods.



This is short compared to the typical lifetime of ultrafine aerosols (hours to ~1 day) (Williams et al., 2002). Thus, the 0.01-0.3 μm sea spray aerosol emission flux from leads could be substantially enhanced under the high wind conditions during both the observed ($U_{9.7m}$ = 11±2 m s$^{-1}$) and falsely predicted (10±1 m s$^{-1}$) BLSN periods, compared to moderate (6±1 m s$^{-1}$) and low wind (3±1 m s$^{-1}$) periods, consistent with the observed increase in particle number concentrations (Fig. 3).


From the OPS measurements of larger particles, for each size bin ranging from 1 to 4 μm $d_o$ (6 size bins), the average particle number concentrations during observed BLSN periods were 67-89% (average 82%) lower than during moderate wind conditions ($p < 0.05$), and 73-89% (average 82%) lower than during low wind periods ($p < 0.05$). Compared to falsely predicted BLSN periods, the average particle number concentrations during observed BLSN periods were 57-63% (average 59%) lower ($p < 0.05$) for 1-2 μm $d_o$ (3 size bins), but not significantly different ($p > 0.05$) for 2-4 μm $d_o$ (3 size bins). This suggests a net removal of 1-4 μm $d_o$ particles by blowing snow. Notably, Pomeroy et al. (1991) suggested that blowing snow can scavenge "micron-sized" SSA via electrophoretic attraction. The scavenging of ambient particles by blowing snow through collisions has not been considered in previous modeling studies (e.g., Huang and Jaeglé, 2017; Huang et al., 2018; Yang et al., 2010; 2019). The collision efficiency between blowing snow and other ambient particles depends on the sizes of these particles (typically on the order of 100 μm and < 10 μm, respectively), with the impact of Brownian motion decreasing with ambient particle size and the contributions of interception and inertial impaction increasing with ambient particle size and mass, respectively (Croft et al., 2009; Seinfeld and Pandis, 2016). Assuming a blowing snow particle diameter on the order of 100 μm, which is the average blowing snow particle diameter observed by Frey et al. (2020) over Antarctic sea ice during winter, the collision efficiency would be lowest with 0.1-1 μm particles and increase for particle diameters both below 0.1 μm (Brownian diffusion) and above 1 μm (interception and inertial impaction) (Croft et al., 2009; Seinfeld and Pandis, 2016). The decrease in the number concentrations of 1-4 μm $d_o$ particles during observed BLSN periods could be explained by greater scavenging by blowing snow than particle production (from open leads and/or blowing snow) in this size range. For 4-5 μm $d_o$ (1 size bin), the average particle number concentrations during observed BLSN periods were not significantly different ($p > 0.05$) from moderate and low wind periods. The enhancement of 0.01-0.06 μm $d_m$ particles during observed BLSN periods (Fig. 3a), as discussed above, suggests more production (by open leads and/or blowing snow) than scavenging (by blowing snow) for these particles.

### 3.3. Chemical composition of atmospheric particles and tundra snowpack

The production of sea spray aerosols and presence of blowing snow both increase with wind speed (Monahan and O'Muircheartaigh, 1980; Nilsson et al., 2001; Yang et al., 2008; Frey et al., 2020). Measurement of the chemical composition of atmospheric particles during different wind conditions aids in source attribution of SSA particles and has been employed in previous Antarctic (Frey et al., 2020; Giordano et al., 2017; Hara et al., 2018; Wagenbach et al., 1998) and Arctic (Hara et al., 2017; Kirpes et al., 2019) studies investigating possible BLSN particle production. Here, we compare bulk (average) submicron ($d_a < 1.1$ μm) and supermicron ($d_a$ 1.1-10 μm) atmospheric particle inorganic ion composition





during low wind speed conditions (Section 3.3.1), moderate wind speed periods (Section 3.3.2), and observed and falsely
predicted BLSN periods (Section 3.3.3). In addition, for selected samples collected during moderate wind and observed
BLSN periods, the chemical composition of individual atmospheric particles ($d_a$ 0.07-0.40 μm and 0.40-2.80 μm) were
measured using CCSEM-EDX, following the previous studies by Kirpes et al. (2018; 2019). As discussed below, the
submicron and supermicron particle inorganic ion and single particle composition shows contribution of local SSA
production (primarily $d_a$ >0.4 μm) during observed BLSN periods, as well as during moderate wind speed periods. Therefore,
in Section 3.3.4, we use ion and elemental ratios to investigate the likely SSA sources during observed BLSN periods
through comparisons to the chemical composition of local tundra surface snowpack (measured in this work), as well as
previous measurements of sea ice surface snowpack (Krnavek et al., 2012), blowing snow (Jacobi et al., 2012), frost flowers
(Douglas et al., 2012), and seawater (Millero et al., 2008).

### 3.3.1 Low wind speed periods

During low wind speed conditions ($U_{9.7m}$ < 4 m s$^{-1}$), blowing snow was neither predicted ($E4$) nor observed (Fig. 2), as
discussed in Section 3.1. Open leads are not expected to be an efficient source of sea spray aerosols at such low wind speeds
(Nilsson et al., 2001; Held et al., 2011), though there could be minor contributions from non-wave-breaking lead-based
bubble bursting sources (Nilsson et al., 2001; Willis et al., 2018). Thus, the atmospheric particle abundance in Utqiaġvik
during low wind periods is expected to be most affected by long-range transport of aerosols contributing to springtime Arctic
haze (Shaw, 1982; Frossard et al., 2011). During the low wind periods in April 3 – May 7, 2016, the elevated average
submicron ($d_a$ < 1.1 μm) particle sulfate concentration (0.7±0.5 μg m$^{-3}$) (Fig. 4), and corresponding average non-sea-salt
sulfate mole fraction ($f_{nssSO_4^{2-}} = \frac{\text{Total SO}_4^{2-} - 0.06\text{N}^+}{\text{Total SO}_4^{2-}}$, Quinn et al. (2002)) of 95±2%, are consistent with the springtime Arctic
haze observed in Utqiaġvik in previous studies (Quinn et al., 2002; 2007). Unfortunately, available bulk supermicron particle
samples primarily corresponded to elevated wind speed periods and did not provide an opportunity for focused low wind
speed period investigation.



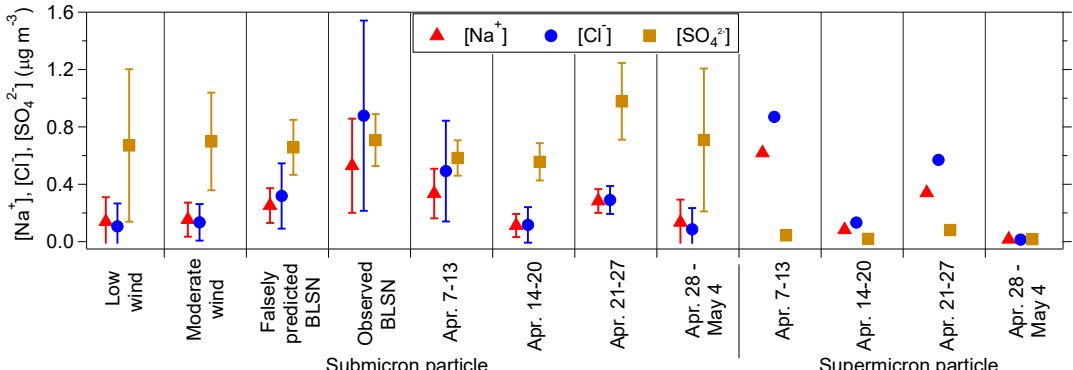

**Figure 4.** Average (± 1σ) [Na⁺], [Cl⁻], and [SO₄²⁻] bulk mass concentrations for daily submicron (<1.1 μm $d_a$) particle samples during low wind, moderate wind, falsely predicted BLSN, and observed BLSN periods, and for multi-day supermicron (1.1-10 μm $d_a$) particles collected on April 7-13 (65% observed BLSN, 35% falsely predicted BLSN), April 14-20 (9% observed BLSN, 29% falsely predicted BLSN, 58% moderate wind, 4% low wind), April 21-27 (3% observed BLSN, 23% falsely predicted BLSN, 47% moderate wind, 27% low wind), and April 28-May 4 (66% moderate wind, 34% low wind). The submicron particle mass concentrations (average ± 1σ) during the four supermicron particle collection periods are also shown for comparison. The corresponding values for these ions, as well as [Ca²⁺], [Mg²⁺], [NH₄⁺], [K⁺], [NO₃⁻], [Br⁻], methanesulfonate [CH₃SO₃⁻] and oxalate (C₂O₄²⁻), can be found in Tables S1 and S2.

During these low wind speed periods ($U_{9.7m}$ < 4 m s⁻¹), the average submicron particle [Cl⁻]/[Na⁺] was 0.4±0.2 mol mol⁻¹ (Fig. 5), much lower than that of seawater (1.2 mol mol⁻¹; Millero et al., 2008), consistent with chloride-depleted and aged SSA. At the same location, May et al. (2016) found submicron particles experienced at least 25% chloride loss relative to seawater during ~80% of the year-round low wind speed periods between 2006 and 2009. In this study, the submicron particle [K⁺]/[Na⁺] (average 0.15±0.08) and [Ca²⁺]/[Na⁺] (average 0.08±0.03) molar ratios during low wind periods (Table S2) were significantly higher than that of seawater (0.02 for both [K⁺]/[Na⁺] and [Ca²⁺]/[Na⁺]; Millero et al., 2008) (Fig. 5), likely due primarily to contributions from long-range transport of biomass burning aerosols and dust (Quinn et al., 2002; 2007; Shaw, 1982; Frossard et al., 2011; Formenti et al., 2003; Ryu et al., 2004).




**Figure 5.** Average (± 1σ) [Cl⁻]/[Na⁺], [SO₄²⁻]/[Na⁺], [Ca²⁺][Na⁺], and [Mg²⁺]/[Na⁺] molar ratios for submicron particles and local tundra surface snowpack for the categories of low wind, moderate wind, falsely predicted BLSN, and observed BLSN periods, and for multi-day supermicron particles collected on April 7-13 (65% observed BLSN, 35% falsely predicted BLSN), April 14-20 (9% observed BLSN, 29% falsely predicted BLSN, 58% moderate wind, 4% low wind), April 21-27





(3% observed BLSN, 23% falsely predicted BLSN, 47% moderate wind, 27% low wind), and April 28-May 4 (66%
moderate wind, 34% low wind). The submicron particle ion ratios (average ± 1σ) during the four supermicron particle
collection periods are shown for comparison. Comparison is also shown to previous measurements of sea ice surface
snowpack (Krnavek et al., 2012), blowing snow (Jacobi et al., 2012), frost flowers (Douglas et al., 2012), and seawater
(Millero et al., 2008).  The corresponding values in this figure can be found in Table S3.

**3.3.2 Moderate wind speed periods**

To investigate particle sources and aging processes during moderate wind speed ($4 \text{ m s}^{-1} \leq U_{9.7m} < U_t$) periods, individual
particles, collected on April 15 19:00 AKDT – April 16 07:00 ($U_{9.7m}$ range 6-8 m s$^{-1}$, average 7 m s$^{-1}$) and April 16 08:00–
20:00 ($U_{9.7m}$ range 4-5 m s$^{-1}$, average 5 m s$^{-1}$), were analyzed by CCSEM-EDX. For $d_a$ 0.07-0.4 μm, seven individual particle
types, including fresh sea spray aerosols, partially aged sea spray aerosols, organic+sulfate aerosols, mineral dust, soot,
potassium+sulfur, and fly ash, were identified (Figures 6 and S4). For $d_a$ 0.4-2.8 μm stage, three particle types were
identified – fresh sea spray aerosol, potassium chloride, and mineral dust; the number fractions of these particle types varied
little with diameter, so their average percentages (± 95% confidence intervals) are reported. These individual particle data aid
in the interpretation of the submicron (<1.1 μm $d_a$) and supermicron (1.1-10 μm $d_a$) inorganic ion mass concentration data
(Fig. 4; Tables S1 and S2). While bulk submicron particle samples were collected on a daily basis over the entire
observational period, the supermicron particle samples collected from April 7 to May 4, 2016 each covered 6-7 days
encompassing multiple wind speed periods. For the bulk supermicron particles, we focus on a sample collected from April
28 to May 4, 2016 that covered NBLSN wind periods (66% moderate wind, 34% low wind; average $U_{9.7m}$=3±1 m s$^{-1}$). In the
following discussion, we focus on the main particle types (organic+sulfate and sea spray aerosol particles), with discussion
of the minor particle types in the Supplementary Information (S1.1).


Organic+sulfate aerosol particles, consistent with Arctic haze (Kirpes et al., 2018), constituted 38-69%, by number, across
the 0.09-0.9 μm $d_{pa}$ range ($d_a$ 0.07-0.4 μm impactor stage) (Fig. 6a). Submicron particle organic carbon and sulfate have
previously been observed during springtime at Utqiaġvik and attributed to long-range transport of particles and precursors
from combustion activities in the Northern Hemisphere, followed by subsequent atmospheric processing, that contribute to
Arctic haze (Quinn et al., 2002; Shaw et al., 2010; Frossard et al., 2011; Hiranuma et al., 2013). Previously, Hiranuma et al.
(2013) observed ~90%, by number, of particles smaller than 0.75 μm ($d_{pa}$) to be organic+sulfate particles during a flight
campaign (up to 6 km) over Utqiaġvik in April 2008. This high organic+sulfate aerosol particle number fraction is consistent
with high submicron particle non-sea-salt sulfate mass concentrations ($[SO_4^{2-}]$ = 0.7 μg m$^{-3}$, $f_{nssSO_4^{2-}}$=97%) measured during
April 15-16. Submicron sulfate mass concentrations were similar during the low ($U_{9.7m}$ < 4 m s$^{-1}$) and moderate wind speed
periods (averages of 0.7 ± 0.5 μg m$^{-3}$ and 0.7 ± 0.3 μg m$^{-3}$, respectively; Table S2). Similarly, the majority of supermicron
particle sulfate during April 28 – May 4 (66% moderate wind) was also non-sea-salt sulfate ($f_{nssSO_4^{2-}}$ = 76%), further





showing the Arctic haze influence. In addition to these organic+sulfate particles, a fraction of the non-sea-salt sulfate was found to be internally mixed in the partially aged sea spray aerosol particles, as previously observed during winter in Utqiaġvik (Kirpes et al., 2018).


Sea spray aerosol particles comprised 8-23%, by number, across the 0.09-0.9 µm $d_{pa}$ range ($d_a$ 0.07-0.4 µm) during April 15-16, with the majority of these individual particles being classified as partially aged (partially chloride-depleted) (Fig. 6a). The average Cl/Na and S/Na molar ratios of these individual sea spray aerosol particles were 0.2±0.1 (95% confidence interval) and 1.0±0.3, respectively, for the April 15-16 sample, and 0.07±0.04 and 1.5±0.7, respectively, for the April 16

sample (Table 3). For comparison, the Cl/Na and S/Na molar ratios in seawater are 1.16 and 0.06, respectively (Millero et al., 2008), showing the significant depletion of chlorine and enrichment of sulfur in the measured sea spray aerosol particles, consistent with the expected aging during transport of submicron sea spray aerosol particles (Kirpes et al., 2018).

These individual particle results are consistent with the average bulk submicron particle [Cl⁻]/[Na⁺] and [SO₄²⁻]/[Na⁺] molar

ratios of 0.5±0.2 and 1.4±0.6 (chloride depletion and sulfate enrichment relative to seawater) during moderate wind periods (Fig. 5), consistent with previous observations at Utqiaġvik (May et al., 2016). Note that both submicron particle [Na⁺] and [Cl⁻] were not significantly different ($p > 0.05$) during moderate wind (4 m s⁻¹ ≤ $U_{9.7m}$ < $U_t$; [Na⁺] = 0.2±0.1 µg m⁻³ and [Cl⁻] = 0.1±0.1 µg m⁻³) and low wind periods ($U_{9.7m}$ < 4 m s⁻¹; [Na⁺] = 0.1±0.2 µg m⁻³ and [Cl⁻] = 0.1±0.2 µg m⁻³) (Fig. 4). Together, these individual sea spray aerosol particle composition and bulk submicron particle mass concentration results are

consistent with transported, rather than locally produced, sea spray aerosol dominating the submicron sea salt aerosol during both low and moderate wind periods.

For the 0.4-2.8 µm $d_a$ stage, the individual sea spray aerosol particles accounted for 30±9%, on average by number, during April 15-16. The average Cl/Na and S/Na molar ratios for these sea spray aerosol particles were 0.93±0.02 and 0.03±0.01,

respectively, for the April 15-16 sample, and 0.77±0.09 and 0.08±0.06, respectively, for the April 16 sample (Table 3). For comparison, the Cl/Na and S/Na molar ratios in seawater are 1.16 and 0.06, respectively (Millero et al., 2008). Therefore, the observed ratios demonstrate significantly less chlorine depletion and no significant sulfur enrichment for sea spray aerosol particles collected on the 0.4-2.8 µm $d_a$ stage, in contrast to the 0.08-0.4 µm $d_a$ sea spray aerosol particles, discussed above, that were more aged. The composition of the larger particles is consistent with locally-produced sea spray aerosols that have

undergone minimal atmospheric processing. Note that Mg was enriched (average Mg/Na molar ratios of 0.4-1.2, Table 3), compared to seawater (0.11; Millero et al., 2008), in the individual sea spray aerosol particles on both the 0.07-0.4 µm $d_a$ and 0.4-2.8 µm $d_a$ stages. This divalent cation binds with organic matter (exopolymeric substances; Krembs et al., 2002) within the sea surface microlayer, resulting in magnesium-enriched organic coatings on the sea spray aerosol (Jayarathne et al., 2016; Bertram et al., 2018; Kirpes et al., 2019).






For the April 28 – May 4 bulk supermicron particle sample, sodium and chloride concentrations (0.017 μg m⁻³ and 0.014 μg m⁻³, respectively) were seven and five times, respectively, lower than the average submicron particle sodium and chloride concentrations (0.1±0.2 μg m⁻³ and 0.1±0.1 μg m⁻³, respectively) during the same period (Fig. 4). The [Cl⁻]/[Na⁺] ratios were 0.6 mol mol⁻¹ and 0.3±0.2 mol mol⁻¹ for supermicron and submicron particles, respectively, during this period (April 28 –

May 4) (Fig. 5). This further confirms that the supermicron SSA were less chloride-depleted and less aged than submicron SSA during this period (Fig. 5), consistent with the expected shorter atmospheric lifetime, larger chloride reservoir, and smaller surface area to volume ratios for the supermicron aerosols (Leck et al., 2002; Williams et al., 2002; Bondy et al., 2017).

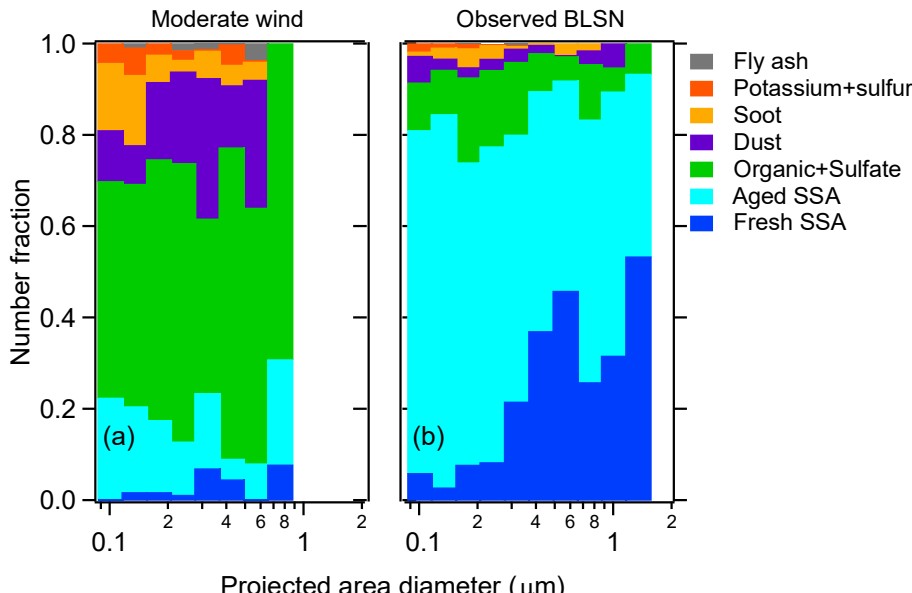


**Figure 6.** Number fractions of seven individual particle types identified by CCSEM-EDX analysis of stage 3 ($d_a$ 0.07-0.4 μm) samples collected during (a) two moderate wind periods (April 15 19:00 – April 16 07:00 AKDT and April 16 08:00–20:00), and (b) two observed blowing snow periods (April 6 07:00–19:00 and April 6 19:00 – April 7 07:00). SSA refers to sea spray aerosol. All particles were collected on the 0.07-0.4 μm $d_a$ impactor stage; projected area diameters, measured by

SEM are shown. Size-resolved counts of particles analyzed by CCSEM-EDX per sample are shown in Figure S3, and representative SEM images and EDX spectra of the particle types are provided in Figure S4.

**Table 3.** Single particle elemental ratios (average ± 95% confidence intervals) of sea spray aerosol particles identified by CCSEM-EDX during (a) two moderate wind periods (April 15 19:00 – April 16 07:00 and April 16 08:00–20:00) and (b)


two observed blowing snow periods (April 6 07:00–19:00 AKDT and April 6 19:00 – April 7 07:00).

| Single particle analysis | Wind category | Impactor stage | Cl/Na (mol mol$^{-1}$) | S/Na (mol mol$^{-1}$) | Mg/Na (mol mol$^{-1}$) | Ca/Na (mol mol$^{-1}$) |
|---|---|---|---|---|---|---|
| April 15 19:00–April 16 07:00 | Moderate | 0.07-0.4 µm $d_a$ | $0.2 \pm 0.1$ | $1.0 \pm 0.3$ | $0.6 \pm 0.3$ | $0.2 \pm 0.2$ |
| | | 0.4-2.8 µm $d_a$ | $0.93 \pm 0.02$ | $0.03 \pm 0.01$ | $0.40 \pm 0.08$ | $0.06 \pm 0.02$ |
| April 16 08:00–20:00 | Moderate | 0.07-0.4 µm $d_a$ | $0.07 \pm 0.04$ | $1.5 \pm 0.7$ | $0.5 \pm 0.2$ | $0.1 \pm 0.1$ |
| | | 0.4-2.8 µm $d_a$ | $0.77 \pm 0.09$ | $0.08 \pm 0.06$ | $1.2 \pm 0.4$ | $0.23 \pm 0.08$ |
| April 6 07:00–19:00 | Observed BLSN | 0.07-0.4 µm $d_a$ | $0.14 \pm 0.02$ | $1.6 \pm 0.1$ | $0.06 \pm 0.01$ | $0.05 \pm 0.05$ |
| | | 0.4-2.8 µm $d_a$ | $0.92 \pm 0.01$ | $0.04 \pm 0.01$ | $0.34 \pm 0.03$ | $0.05 \pm 0.01$ |
| April 6 19:00–April 7 07:00 | Observed BLSN | 0.07-0.4 µm $d_a$ | $0.34 \pm 0.07$ | $2.6 \pm 0.2$ | $0.17 \pm 0.02$ | $0.04 \pm 0.01$ |

### 3.3.3 Observed and falsely predicted blowing snow periods

The same seven individual particle types (fresh sea spray aerosols, partially aged sea spray aerosols, organic+sulfate aerosols, mineral dust, soot, potassium+sulfur, and fly ash), observed during the moderate wind speed period, were also present on the

0.07-0.4 µm $d_a$ impactor stage during an observed BLSN period: April 6 07:00–19:00 AKDT ($U_{9.7m}$ range 11-13 m s$^{-1}$, average 12 m s$^{-1}$) and April 6 19:00 – April 7 07:00) ($U_{9.7m}$ range 9-12 m s$^{-1}$, average 11 m s$^{-1}$) (Fig. 6b). Fresh and partially aged sea spray aerosol particles constituted 74-93%, by number, across the 0.09-1.5 µm $d_{pa}$ range, during this observed BLSN period (Fig. 6b). In comparison, during the moderate wind period examined (April 15-16), fresh and partially aged sea spray aerosol particles comprised only 8-23%, by number, across the 0.09-0.9 µm $d_{pa}$ range (Fig. 6a). In contrast, the number

fractions of organic+sulfate aerosol particles were lower during this observed BLSN period (5-19%, by number, across 0.09-1.5 µm $d_{pa}$; Fig. 6b), compared to that during the moderate wind period of April 15-16 (38-69%, by number, across 0.09-0.9 µm $d_{pa}$; Fig. 6a). Similar to the moderate wind periods, the majority of submicron particle sulfate during observed and falsely predicted BLSN periods was non-sea-salt sulfate, with average $f_{nssSO_4^{2-}}$ of 82±9% and 90±5%, respectively, indicative of Arctic haze influence. Both the submicron particle total sulfate and non-sea-salt sulfate concentrations were not significantly

different ($p > 0.05$) during observed BLSN periods ([SO$_4^{2-}$]=0.7±0.2 µg m$^{-3}$; [SO$_4^{2-}$]$_{nss}$=0.6±0.2 µg m$^{-3}$) and low wind periods ([SO$_4^{2-}$]=0.7±0.5 µg m$^{-3}$; [SO$_4^{2-}$]$_{nss}$=0.6±0.5 µg m$^{-3}$) (Fig. 4), suggesting the contribution of long-range transported secondary sulfate was similar during these two period types.

Observed BLSN periods featured average submicron particle [Na$^+$] of 0.5±0.3 µg m$^{-3}$, compared to 0.2±0.1 µg m$^{-3}$ during

moderate wind periods and 0.1±0.2 µg m$^{-3}$ during low wind periods (Fig. 4.) Similarly, [Cl$^-$] increased from 0.1±0.1 µg m$^{-3}$





and 0.1±0.2 μg m$^{-3}$ during moderate and low wind periods, respectively, to 0.9±0.7 μg m$^{-3}$ during observed BLSN periods (Fig. 4). This is consistent with a submicron sea salt source during observed BLSN periods. The increased submicron particle [Na$^+$] and [Cl$^-$] concentrations during observed BLSN periods, compared to moderate wind periods (Fig. 4), is consistent with the measured increased relative contribution from individual sea spray aerosol particles.


Among all of the measured individual sea spray aerosol particles collected on the 0.07-0.4 μm $d_a$ impactor stage during the April 6-7 observed BLSN period, fresh (nascent) sea spray aerosol particles comprised 3-11% and 27-57%, by number, across the 0.1-0.3 μm $d_{pa}$ and 0.3-1.3 μm $d_{pa}$ ranges, respectively (Fig. 6b). For the larger (0.4-2.8 μm $d_a$) impactor stage, only results from the April 6 07:00–19:00 BLSN period are reported because the particle loading on the April 6 19:00 –

April 7 07:00 substrate was insufficient for CCSEM-EDX analysis, consistent with reduced particle number concentrations in this diameter range during observed BLSN periods (Section 3.2). For the April 6 observed BLSN period, fresh sea spray aerosol particles comprised 52±8%, by number, of the individual particles collected on the 0.4-2.8 μm $d_a$ impactor stage during the April 6 observed BLSN period, compared to 30±9% observed during the moderate wind period (April 15-16). As discussed in Section 3.3.2, these percentages represent upper limits for the fresh sea spray aerosol content for particles

collected on the 0.4-2.8 μm $d_a$ impactor stage due to challenges in detecting organic particles on the aluminum substrate. Together with the trend observed for the 0.07-0.4 μm $d_a$ impactor stage, this shows an increased relative contribution of locally produced sea spray aerosol with increasing particle diameter, as previously observed by Kirpes et al. (2018).

Aged sea spray aerosol particles (Cl-depleted, S-enriched) (Kirpes et al., 2018) comprised 89-97% and 43-73%, by number,

across the 0.1-0.3 μm $d_{pa}$ and 0.3-1.3 μm $d_{pa}$ ranges, of the measured individual sea spray aerosol particles during the April 6-7 observed BLSN period. In comparison to this 0.07-0.4 μm $d_a$ impactor stage, only fresh sea spray aerosol particles were detected on the 0.4-2.8 μm $d_a$ impactor stage. This size-dependent trend is consistent with increasing aging for smaller sea spray aerosols, due to their smaller Cl reservoir, greater surface area to volume ratio, and longer atmospheric lifetime (Leck et al., 2002; Williams et al., 2002; Bondy et al., 2017; Kirpes et al., 2018).


For the total individual sea spray aerosol particles collected on the 0.07-0.4 μm $d_a$ impactor stage, the average Cl/Na and S/Na molar ratios were 0.14±0.02 and 1.6±0.1, respectively, for the April 6 sample, and 0.34±0.07 and 2.6±0.2, respectively, for the April 6 – 7 sample (Table 3). These ratios are indicative of chloride-depleted and sulfate-enriched aged sea spray aerosol, on average (Kirpes et al., 2018), since the Cl/Na and S/Na molar ratios in seawater are 1.16 and 0.06, respectively

(Millero et al., 2008). The bulk submicron ($d_a$ < 1.1 μm) particle [Cl$^-$]/[Na$^+$] on April 6 was 1.2 mol mol$^{-1}$, which is higher than that observed for the 0.07-0.4 μm $d_a$ individual sea spray aerosol (0.14±0.02), suggesting that most of the submicron fresh SSA mass was >0.4 μm $d_a$, as expected based on previous measurements of sea spray aerosol size distributions, including those produced from sea ice leads (Nilsson et al., 2001; Feng et al., 2017). Consistent with greater local sea salt contribution to the upper submicron range (0.4-1.1 μm $d_a$), the bulk submicron particle [SO$_4^{2-}$]/[Na$^+$] molar ratio on April 6





(0.2) was lower than that observed for the <0.4 µm ($d_a$) individual sea spray aerosol particles (S/Na during April 6: 1.6±0.1,

April 6 – 7: 2.6±0.2, Table 3). Therefore, these observations further confirm that the increased submicron SSA mass

concentrations during observed BLSN periods were primarily at greater than 0.4 µm ($d_a$), both in terms of number and mass.

In comparison, for the individual sea spray aerosol collected on the 0.4-2.8 µm $d_a$ impactor stage on April 6, the average

Cl/Na and S/Na molar ratios were 0.92 ± 0.01 and 0.04 ± 0.01, respectively. This result further supports the trend of

decreased aging, and increasing local source contribution, with increasing diameter.

Overall, the bulk submicron ($d_a$ < 1.1 µm) mass concentrations showed a higher average $[Cl^-]/[Na^+]$ molar ratio during

observed BLSN periods (0.9±0.3), compared to averages during moderate (0.5±0.2) and low (0.4±0.2) wind periods (Fig. 5),

is indicative of less chloride depletion and the dominant local sea salt aerosol production, considering the seawater

$[Cl^-]/[Na^+]$ molar ratio of 1.2 (Millero et al., 2008). In comparison, the submicron particle $[Na^+]$, $[Cl^-]$, and $[Cl^-]/[Na^+]$

during falsely predicted BLSN periods were 0.3±0.1 µg m$^{-3}$, 0.3±0.2 µg m$^{-3}$, and 0.8±0.2 mol mol$^{-1}$, respectively, between

the values for observed BLSN and moderate wind periods.

The increased $[Na^+]$, $[Cl^-]$, and $[Cl^-]/[Na^+]$ during BLSN wind periods, compared to NBLSN wind periods, were also

observed for supermicron particles (Figures 4 and 5). The April 7-13 supermicron particle sampling period covered BLSN

wind periods (average $U_{9.7m}$ = 11±2 m s$^{-1}$; 65% observed BLSN, 35% falsely predicted BLSN). An additional two

supermicron samples covered a wide range of wind speeds: April 14-20 ($U_{9.7m}$ = 4-13 m s$^{-1}$, average 7 m s$^{-1}$, 9% observed

BLSN, 29% falsely predicted BLSN, 58% moderate wind, 4% low wind) and April 21-27 ($U_{9.7m}$ = 0-13 m s$^{-1}$, average 6 m s$^{-1}$,

3% observed BLSN, 23% falsely predicted BLSN, 47% moderate wind, 27% low wind). During April 7-13 (BLSN wind),

supermicron particle $[Na^+]$ (0.619 µg m$^{-3}$) and $[Cl^-]$ (0.870 µg m$^{-3}$) were 35 and 61 times higher, respectively, than during

April 28 – May 4 (NBLSN wind, 0.017 µg m$^{-3}$ for $[Na^+]$ and 0.014 µg m$^{-3}$ for $[Cl^-]$), consistent with a significant

supermicron SSA source during April 7-13. The $[Na^+]$, $[Cl^-]$, and $[Cl^-]/[Na^+]$ were also elevated for the submicron particle

samples during April 7-13, compared to April 28 – May 4 (Figures 4 and 5). In comparison, the supermicron particle $[Na^+]$

and $[Cl^-]$ during April 14-20 and April 21-27, which covered both BLSN and NBLSN wind periods, were in between the

April 7-13 (BLSN wind) and April 28 – May 4 (NBSLN wind) periods (Fig. 4). The supermicron particle $[Cl^-]/[Na^+]$ was

0.9 mol mol$^{-1}$ during April 7-13 (BLSN wind), indicative of limited chloride depletion (seawater $[Cl^-]/[Na^+]$ = 1.2; Millero et

al., 2008), similar to the submicron particle molar ratio during the same period (0.9±0.2). Similarly, the supermicron particle

$[Cl^-]/[Na^+]$ molar ratios for the April 14-20 and April 21-27 samples were 1.0 and 1.1 (Fig. 5), respectively, which are higher

than the submicron particle ratios during the same periods (0.6±0.2 and 0.7±0.1). The minor chloride depletion of all three

supermicron particles samples collected during April 7-27 is consistent with the individual particle composition and

contribution of local supermicron sea salt aerosol production during the moderate and BLSN wind speeds.

For the April 7-13 supermicron particle sample during observed BLSN, the $[SO_4^{2-}]/[Na^+]$ molar ratio was 0.017, which is





lower than that of seawater (0.06 mol mol$^{-1}$; Millero et al., 2008) (Fig. 5). Taken alone, this would be suggestive of a sulfur-
depleted supermicron SSA source. However, the average S/Na molar ratio of the 0.4-2.8 μm $d_a$ individual sea spray aerosol
particles during the observed BLSN period on April 6 was $0.04 \pm 0.01$, similar to the ratio of $0.03 \pm 0.01$ observed during the
moderate wind speed period on April 15-16 (Table 3). This suggests that the bulk ratio is an underestimate when considering
the distribution of sulfate and sodium across the atmospheric particle population. In comparison, the $[SO_4^{2-}]/[Na^+]$ molar
ratios were 0.056 and 0.057, respectively, for the April 14-20 and April 21-27 supermicron particle samples that experienced
both BLSN and NBLSN wind speeds (Fig. 5). Therefore, the low bulk $[SO_4^{2-}]/[Na^+]$ ratio for the April 7-13 supermicron
particle sample can be explained by the increased $[Na^+]$, from sea spray aerosol generation, at the higher wind speeds, with
similar $[SO_4^{2-}]$, from haze aerosol, across the entire wind speed range (Fig. 4). This observation highlights that bulk ratios
are not always representative of the individual particle population that includes multiple particle types/sources and that
caution must be taken in their interpretation.

**3.3.4 Sea salt aerosol sources during observed blowing snow**

The above analysis of submicron and supermicron particle inorganic ion and single particle composition shows contribution
of local SSA production (primarily >0.4 μm) during observed BLSN periods. Open leads, exposing Arctic Ocean water
directly to the atmosphere, have been observed in the field to serve as an efficient source of SSA at high wind speeds
(Nilsson et al., 2001; Leck et al., 2002; May et al., 2016; Kirpes et al., 2019). In addition, blowing snow sublimation has
been proposed as a potential source of SSA at high wind speeds in the Polar regions by modeling (Yang et al., 2008; 2019;
Huang et al., 2018) and field (Giordano et al., 2018; Frey et al., 2020) studies. While previously proposed as a source of SSA,
frost flowers are mechanically rigid and difficult to break and produce ambient aerosols even under strong winds, as shown
in both laboratory (Roscoe et al., 2011; Yang et al., 2017) and field (Domine et al., 2005; Alvarez-Aviles et al., 2008) studies.
Here, we follow previous observational studies to use ion and elemental ratios to discern the likely SSA source(s) (Hara et al.,
2017; Kirpes et al., 2019; Quinn et al., 2002). As discussed in detail below, both submicron and supermicron inorganic SSA
composition is consistent with sea spray aerosol production from open leads, similar to during moderate wind speeds, but at
higher concentrations due to the production flux dependence on wind speed (Nilsson et al., 2001). If blowing snow
sublimation contributed to the observed SSA population, the snow being sublimed must have been primarily impacted by sea
spray aerosol deposition, rather than upward brine migration through the snowpack.


During observed BLSN periods, the average $[Cl^-]/[Na^+]$ and $[K^+]/[Na^+]$ molar ratios of local tundra surface snowpack were
$1.1\pm0.1$ and $0.019\pm0.004$, respectively, similar to those of seawater (1.2 and 0.02, respectively; Millero et al., 2008) (Fig. 5).
This is consistent with SSA deposition on the tundra snowpack (Simpson et al., 2005). These observations for the tundra
snowpack are also consistent with previous observations for springtime Beaufort Sea first-year sea ice surface snowpack
($[Cl^-]/[Na^+]$ and $[K^+]/[Na^+]$ = 1.2 and 0.02, respectively; Krnavek et al., 2012) and springtime blowing snow collected over
tundra near Utqiaġvik ($1.21\pm0.07$ and $0.022\pm0.003$, respectively; Jacobi et al., 2012) (Fig. 5). The snowpack over sea ice is





influenced by both SSA deposition and upward brine migration (Domine et al., 2004; Peterson et al., 2019). Thus, blowing snow particles lofted from both the local tundra surface snowpack and Beaufort Sea first-year sea ice surface snowpack would be expected to possess [Cl⁻]/[Na⁺] and [K⁺]/[Na⁺] signatures similar to those of seawater. In comparison, during

observed BLSN, the submicron and supermicron particle [Cl⁻]/[Na⁺] molar ratios ($0.9\pm0.3$ and $0.9$, respectively) showed only minor chloride depletion (Fig. 5), consistent with local production. Bulk submicron and supermicron particle [K⁺]/[Na⁺] molar ratios ($0.04\pm0.01$ and $0.03$, respectively) during observed BLSN were 60-120% higher than those of seawater and the snowpack (Fig. 5). Individual particle analyses showed that this bulk observation is not consistent with individual SSA particle composition and is due instead to the presence of externally mixed potassium-rich particles, as shown in Fig. 6 and

discussed in the Supplementary Information.

The average [Ca²⁺]/[Na⁺] molar ratio of the springtime Beaufort Sea first-year sea ice surface snowpack (0.02; Krnavek et al., 2012) is similar to that of seawater (0.02; Millero et al., 2008) (Fig. 5). In comparison, the [Ca²⁺]/[Na⁺] molar ratio of the local tundra surface snowpack during observed BLSN periods was slightly higher ($0.033\pm0.004$), with previously collected

springtime blowing snow over Utqiaġvik tundra even more enriched in calcium ($0.06\pm0.02$; Jacobi et al., 2012). The tundra snow (this study) and blowing snow (Jacobi et al., 2012) composition are consistent with the measured submicron particle [Ca²⁺]/[Na⁺] ($0.04\pm0.01$) during observed BLSN periods and supermicron particle [Ca²⁺]/[Na⁺] ($0.03$) during observed + falsely predicted BLSN periods (April 7-13). These bulk ratios are consistent with the average Ca/Na molar ratios of 0.04-0.05 in the individual sea spray aerosol particles (0.07-2.8 μm $d_a$) during the April 6-7 BLSN event (Table 3), when there

was little influence from dust particles (Fig. 6 and Section S1.1). Similar to calcium, magnesium enrichment, relative to sodium, was also observed for the submicron particles during observed BLSN, supermicron particles during observed + falsely predicted BLSN (April 7-13), the individual 0.07-0.4 μm $d_a$ sea spray aerosol particles collected overnight from April 6 19:00 to April 7 07:00, and the individual 0.4-2.8 μm $d_a$ sea spray aerosol particles, collected during April 6 daytime (Fig. 5, Table 3, and Section S1.3). Among the seven individual particle sampling periods, only the individual 0.07-0.4 μm $d_a$ sea

spray aerosol particles collected on April 6 showed magnesium depletion, as discussed in Section S1.3.

Calcium and magnesium enrichment in the sea spray aerosols is due to binding of the divalent cation with organic matter (exopolymeric substances) (Krembs et al., 2002) in the surface seawater microlayer that leads to organic coatings on sea salt particles during bubble bursting (Jayarathne et al., 2016; Salter et al., 2016; Bertram et al., 2018). Indeed, the individual 0.4-

2.8 μm $d_a$ fresh sea spray aerosol lacking chloride depletion were enriched in carbon (Figure S5). This has been previously observed for wintertime sea spray aerosols produced from leads near Utqiaġvik (Kirpes et al., 2019). Since sea spray aerosol deposition influences the surface snowpack composition above both sea ice and tundra (Domine et al., 2004; Peterson et al., 2019), the calcium enrichment in the tundra snowpack and blowing snow suggest surface snow influence from sea spray aerosol deposition (Krnavek et al., 2012) and resuspension during BLSN conditions.




The bulk submicron particle mass during observed BLSN periods was enriched in sulfate compared to seawater (Fig. 5), and 90±5% of the sulfate was calculated to correspond to secondary non-sea-salt sulfate associated with Arctic haze (Section 3.3.3). This is consistent with the high number fraction of aged (Cl depleted, S enriched) sea spray aerosol (0.07-0.4 µm $d_a$) (Fig. 6), leading to average S/Na ratios of 1.6±0.1 and 2.6±0.2 during observed BLSN (Table 3). In contrast, the bulk supermicron particle $[SO_4^{2-}]/[Na^+]$ during observed BLSN (April 7-13) was 0.017 mol mol$^{-1}$, which is lower than that of seawater (0.06; Millero et al., 2008) and most similar to that of frost flowers (0.016±0.003; Douglas et al., 2012) (Fig. 5). Yet, frost flowers themselves are not expected to produce aerosols (Domine et al., 2005; Alvarez-Aviles et al., 2008; Roscoe et al., 2011; Yang et al., 2017). However, the average S/Na ratio of the 0.4-2.8 µm $d_a$ individual sea spray aerosol particles during the observed BLSN period on April 6 was $0.04 \pm 0.01$, similar to the ratio of $0.03 \pm 0.01$ observed during the moderate wind speed period on April 15-16 (Table 3). This shows that the bulk supermicron $[SO_4^{2-}]/[Na^+]$ ratio is not representative of the individual sea spray aerosol particles.

The average $[SO_4^{2-}]/[Na^+]$ molar ratio of the local tundra surface snowpack during observed BLSN periods was 0.04±0.01, similar to that of springtime Beaufort Sea first-year sea ice surface snowpack (0.04; Krnavek et al., 2012) and springtime blowing snow collected over tundra near Utqiaġvik (0.03±0.01; Jacobi et al., 2012), but lower than that of seawater (0.06; Millero et al., 2008) (Fig. 5). This ratio is similar to the S/Na ratios of the 0.4-2.8 µm $d_a$ individual sea spray aerosol particles during the observed BLSN period on April 6 ($0.04 \pm 0.01$); however, it is also similar to the ratio ($0.03 \pm 0.01$) measured for the moderate wind speed period on April 15-16 (Table 3). Sulfate depletion can result from mirabilite (Na$_2$SO$_4$□10H$_2$O) precipitation below -6.4 °C (Wagenbach et al., 1998; Butler et al., 2016) within brine that influences the composition of the first-year sea ice snowpack. The air temperature at 1.3 m during observed BLSN periods was -14±3 °C (range from -22 °C to -7 °C), supporting mirabilite precipitation. Previously, frost flowers with an average $[SO_4^{2-}]/[Na^+]$ molar ratio of 0.016±0.003 were collected near Utqiaġvik during spring (Douglas et al., 2012). The snowpack (over tundra and Beaufort Sea first-year sea ice) $[SO_4^{2-}]/[Na^+]$ was between that of frost flowers and seawater. Since frost flower and upward brine migration only impact snow above sea ice (Domine et al., 2004), the sulfate signature of the tundra surface snowpack could be due to the transport of nearby (~5 km) coastal sea ice blowing snow to the nearby tundra. Tabler et al. (1990) reported that snow particles can be blown for distances of 3 to 5 km, corresponding to atmospheric residence times of up to 10 min. For the average U$_{9.7m}$ of 11 m s$^{-1}$ during observed BLSN (Table 2), snow particle transport for 5 km would correspond to ~7.5 min.

Overall, the observed minor chloride depletion for particles >0.4 µm $d_a$ points to the influence of locally produced submicron and supermicron sea salt aerosols during observed BLSN. Individual sea spray aerosol particles <0.4 µm $d_a$ were primarily depleted in chloride and enriched in sulfate, consistent with aging during transport and the significant non-sea-salt sulfate mass concentrations. Most notably, calcium enrichment in both submicron and supermicron sea spray aerosol particles, regardless of chloride depletion, is consistent with divalent cation binding with organic coatings obtained during bubble



bursting within open leads, with sea spray aerosol deposition on the surface snowpack likely explaining tundra snowpack calcium enrichment. Therefore, the bulk submicron particle mass composition is consistent with sea spray aerosol produced from bubble bursting within open leads. If blowing snow sublimation contributed to the observed SSA population, the snow being sublimed must have been primarily impacted by sea spray aerosol deposition, rather than upward brine migration through the snowpack.

**4. Conclusions**


This study investigates the impacts of leads and blowing snow on ambient particle number concentrations and chemical composition at a coastal Arctic site (Utqiaġvik, AK) during spring 2016. Our unique dataset consists of blowing snow observations by the local airport and ceilometer, size-resolved aerosol number concentrations from 0.01 to 5 μm, and both bulk and single-particle chemical atmospheric particle composition. The observed BLSN periods, when blowing snow was

both predicted and observed by either the local airport or ceilometer, covered 25% of the five-week study. Only 57% of predicted BLSN periods, based on wind speed and temperature, were observed by the airport or ceilometer. This suggests that the commonly used parameterization based on data from the Canadian Prairies (Li and Pomeroy, 1997) overestimates the occurrence of blowing snow when simulating it simply based on wind speed and air temperature, as previously observed in Antarctica (Gossart et al., 2017). Therefore, atmospheric chemistry modeling studies using this parameterization in the

Polar Regions likely overestimate the frequency of blowing snow, and thus the potential contribution of sea salt aerosols from blowing snow sublimation. Future studies are needed to further investigate blowing snow occurrence and sublimation flux.

The average particle number concentration from 0.01-0.06 μm $d_m$ was on average six times higher during observed BLSN

periods, compared to low wind periods, suggesting ultrafine particle production in the presence of blowing snow and elevated wind speeds. Future chemical composition measurements are required to identify the source(s) of these ultrafine particles. Based on previous sea spray aerosol measurements, it is expected that ultrafine sea spray aerosol production from leads would be enhanced under the high wind and low temperature conditions (Clarke et al., 2006; Nilsson et al., 2001; Zábori et al., 2012). Future measurements are needed of size-resolved sea spray aerosol emission fluxes from leads as a

function of wind speed and fetch.

The 0.06-0.3 μm $d_m$ particle number concentration was on average 67% higher during observed BLSN periods, compared to low wind periods. In this particle size range, a significantly increased relative number contribution from aged (partially chloride-depleted, sulfate-enriched) and fresh/nascent sea spray aerosol was observed during BLSN, with primarily

organic+sulfate particles dominating the low wind periods, as expected for Arctic haze influence. The individual submicron sea spray aerosol particles were enriched in calcium relative to seawater during both moderate and observed BLSN



conditions, consistent with divalent cation binding with organic matter obtained from the sea surface microlayer during bubble bursting (Kirpes et al., 2019; Salter et al 2016, Jayarathne et al., 2016, and Bertram et al 2018). These single-particle observations point to the heterogeneity of the aerosol population with multiple contributing particle sources and aging
processes that are a challenge to disentangle with bulk aerosol composition data alone. Submicron sulfate mass concentrations were similar across all wind speed conditions, while both sodium and chloride mass concentrations increased with increasing wind speed (low to moderate to falsely predicted and observed BLSN). This is consistent with the previously observed increasing emission flux of sea spray aerosol particles, including from leads, upon increasing wind speeds (Nilsson et al., 2001; Leck et al., 2002; Monahan and O'Muircheartaigh, 1980).


To produce sea salt aerosol particles of 0.01-0.3 μm that were enhanced during BLSN in the current work, small blowing snow particles of 0.6-19 μm are needed, when assuming only one sea salt aerosol particle is produced from one blowing snow particle (Yang et al., 2008). Recent atmospheric chemistry modeling studies have assumed multiple sea salt aerosol particles are produced from the sublimation of one larger blowing snow particle (Huang and Jaeglé, 2017; Huang et al.,
2018; Yang et al., 2019), but to our knowledge, there are no experimental studies on this process. In addition, these modeling studies have assumed blowing snow particles contain only sea salt, in contrast to various chemical components in the Arctic snowpack, including organics, dust, nitrate, sulfate, and soot (e.g., Macdonald et al., 2018; Mori et al., 2019; Grannas et al., 2007; Krnavek et al., 2012). Thus, large uncertainties exist when modeling sea salt aerosol production from blowing snow sublimation, and future studies are needed to assess the distribution of the size and number of particles produced during the
sublimation of a blowing snow grain.

While 0.4-1.0 μm $d_o$ particle number concentrations were not enhanced during observed BLSN, differences in particle chemical composition were observed, compared to moderate and low wind periods. Increased submicron particle sodium and chloride mass concentrations were measured for observed BLSN periods, compared to moderate and low wind periods. This
is consistent with the observation of the presence of individual 0.4-2.8 μm fresh (nascent) sea spray aerosol particles, further suggesting local submicron sea salt aerosol production. To investigate the source(s) of the sea salt aerosol, inorganic ion ratios were compared to those of local tundra surface snowpack in this study, as well as sea ice surface snowpack, blowing snow, frost flowers, and seawater in previous studies. Measured submicron particle $[Cl^-]/[Na^+]$ molar ratios were similar to those of seawater and the local tundra snowpack, as well as previous measurements of the first-year sea ice snowpack
(Krnavek et al., 2012) and blowing snow (Jacobi et al., 2012) near Utqiaġvik. Calcium enrichment, relative to seawater, during moderate and observed BLSN periods is consistent with open lead production of sea spray aerosols due to the binding of these divalent cations with organic matter obtained from the sea surface microlayer during bubble bursting (Kirpes et al., 2019; Jayarathne et al., 2016). The similar calcium enrichment observed for the local tundra surface snowpack and previous blowing snow (Jacobi et al., 2012) is likely due to the influence of sea spray aerosol deposition on the surface snowpack, as
it is not expected for upward brine migration or frost flower influence on the snowpack above the sea ice. Therefore, the





measured submicron aerosol composition is consistent with bubble bursting production within open leads that were observed visually and by satellite throughout the whole study.

In contrast to the enhancements at <0.3 µm $d_m$, the average particle number concentration from 1-4 µm $d_o$ was, on average,

82% lower during observed BLSN periods than during low wind periods, suggesting scavenging of particles by blowing snow, a process currently missing in global models, including those that incorporate blowing snow sublimation (e.g. Huang et al., 2018; Yang et al., 2019). This decreased number concentration is despite the increased supermicron sea salt ([Na$^+$] and [Cl$^-$]) mass concentrations observed during the moderate and BLSN periods. Only minor chloride depletion, relative to seawater, was measured, indicating local supermicron sea salt aerosol production. Similar to the submicron particles, the sea

spray aerosol enrichment in Ca$^{2+}$ and Mg$^{2+}$ is consistent with binding of these divalent cations with organic matter that coated the sea salt particles during bubble bursting at the sea surface microlayer (Kirpes et al., 2019; Jayarathne et al., 2016). Therefore, if blowing snow sublimation contributed to the observed submicron and/or supermicron sea salt aerosol populations, the snow being sublimated must have been impacted by sea spray aerosol deposition, rather than upward brine migration through the snowpack. It is important to note that the bulk inorganic ion ratios were not indicative of the

individual sea spray aerosol particle composition due to the contribution of externally mixed sulfate haze aerosol. This observation highlights that bulk ratios are not always representative of the individual particle population (Riemer et al., 2019) and that caution must be taken in their interpretation. This further points to the need for individual atmospheric particle measurements in future studies to more accurately assess source contributions.

The thinning and fracturing sea ice across the Arctic Ocean is expected to be increasing sea spray aerosol production from open leads, with implications for aerosol optical depth and aerosol-cloud interactions (Struthers et al., 2011; Browse et al., 2014). An improved understanding of aerosol emissions from leads is therefore critical to future prediction of the Arctic sea spray aerosol burden. Future observational studies are also needed to further investigate the conditions under which blowing snow occurs and its contribution to aerosol production, as well as its role in scavenging of supermicron particles. Reduced

sea ice extent will provide less sea ice surface area for blowing snow production. However, with decreasing snow depth on Arctic sea ice (Webster et al., 2014), sea ice surface snow salinity is expected to increase, as it depends on both upward brine migration from the sea ice (increasing with decreasing snow depth) and snow surface deposition of sea spray aerosols from nearby open leads (Domine et al., 2004; Peterson et al., 2019).


***Data availability.*** Data for the blowing snow events reported from the Wiley Post-Will Rogers Memorial Airport are available at https://www.wunderground.com/. DOE ARM ceilometer data are available at https://www.arm.gov/capabilities/instruments/ceil, or by contacting A.K. (aaron.kennedy@und.edu). Submicron and supermicron particle inorganic ion data are available by contacting P.K.Q. (patricia.k.quinn@noaa.gov). Sonic anemometer,



SMPS, OPS, CCSEM-EDX, and tundra surface snow ion composition data are available by contacting the corresponding author.

*Author contributions.* QC and KAP prepared the manuscript and led data integration, analysis, and interpretation. PBS, KAP, and JDF designed the PHOXMELT field campaign. QC and KAP conceived this study. JAM and RMK conducted the

CCSEM-EDX single particle analysis and interpretation, with assistance from SC, NNL, and APA. ST and JDF conducted sonic anemometer measurements. ST, ARWR, and SMM conducted SMPS and OPS size-resolved aerosol number concentration measurements. ARWR and SMM collected tundra snow and particle samples and contributed to snow ion concentration measurements, conducted by AJB. NAL and AK provided the ceilometer data analysis and interpretation. LMU and PKQ provided submicron and supermicron particle ion concentration measurements.


*Competing interests.* The authors declare that they have no conflict of interests.

*Acknowledgements.* This study was supported by the National Science Foundation (PLR-1417668, PLR-1417906, PLR-1417914, OPP-2000493, OPP-2000428, OPP-2000403), the National Aeronautics and Space Administration Earth Science Program (NNX14AP44G), the DOE Atmospheric Systems Research program (DE-SC0019392), and a Sloan Research

Fellowship to K. A. Pratt. LMU acknowledges the Cooperative Institute for Climate, Ocean, & Ecosystem Studies (CIOCES) under NOAA Cooperative Agreement NA20OAR4320271 (Contribution No. 2022-1206). QC acknowledges the Hong Kong General Research Fund (15223221) for effort during manuscript revisions. We thank the Ukpeaġvik Iñupiat Corporation - Science and Polar Field Services, as well as Dandan Wei and Jesus Ruiz-Plancarte from Pennsylvania State University, for logistical support during this field campaign. We thank NOAA ESRL Barrow Observatory for CPC aerosol

number concentration data and DOE ARM Barrow Observatory for ceilometer data. CCSEM-EDX analyses were performed at the Environmental Molecular Sciences Laboratory (EMSL), a national scientific user facility located at the Pacific Northwest National Laboratory (PNNL) and sponsored by the Office of Biological and Environmental Research of the US Department of Energy (DOE). PNNL is operated for DOE by Battelle Memorial Institute under contract no. DE-AC06-76RL0 1830. P.K.Q acknowledges PMEL contribution number 5174.

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
