# Peer review of "Atmospheric particle abundance and sea salt aerosol observations in the springtime Arctic: a focus on blowing snow and leads"

_Atmospheric Chemistry and Physics, 2022_

## Author Comment (AC1)

We thank the reviewers for giving helpful comments (shown in gray font below) on the manuscript. Please see our point-to-point responses (shown in blue font below), with reference to the line numbers in the track changes version of the manuscript.

**Reviewer #1:**

Review of "Atmospheric particle abundance and sea salt aerosol observations in the springtime Arctic: a focus on blowing snow and leads"

Chen et al. provide an interesting case study of aerosol and sea salt observations taken on the North Slope of Alaska. The results are then used to evaluate parameterizations for predicting blowing snow as well as the sources of SSA (e.g., sublimation of blowing snow or sea ice leads). The manuscript is well-written, although it is too long in certain sections and it might benefit from some cuts to the lengthy discussions of particle composition, providing more of a summary and leaving the details in the tables and figures.

Thank you. As suggested and discussed below, we have significantly shortened Sections 3.3.2, 3.3.3, and 3.3.4, moving this content instead to the Supplemental Information.

Overall, the manuscript is of high technical quality, but certain key measurements, particularly the SEM-EDX analyses, have only been taken on a small number of samples, which decreases confidence in the representativity of the results. The best solution would be to analyze more samples, but I realize that this might not be feasible given the nature of field work in the Arctic. An alternative would be to modify the discussion and conclusions to better acknowledge these limitations. I support publication in ACP once my comments have been considered.

Thank you. Please see specific responses below regarding the SEM-EDX analyses.

Specific comments.

Blowing snow observations: Is there anyway to evaluate if the blowing snow observations, which were taken at other sites, reflect the conditions at Utqiagvik? The distance between the airport or the DOE ARM site and the Utqiagvik site is approximately 5 km. Is it possible that blowing snow occurs at different times at the different sites? Could this explain the large number of periods that blowing snow is falsely predicted? It is very important to provide further details about the topography, vegetation (probably not relevant for an Arctic site) and local meteorological conditions to evaluate the potential differences between the sites in this respect.

We added a topography map in the supplement (Fig. S6, below) showing the elevation of the local area, which includes our field site, the airport, and the ARM site in Utqiaġvik. The elevation difference between these three sites is within 5 m. In addition, the surface type within the area is mainly flat tundra. There is no significant difference in meteorological parameters; generally the measurements were within 0.3 m s$^{-1}$ for wind speed, 1 °C for air temperature, and 1% for relative humidity between these 3 sites, as now shown in Figure S7 (below). Therefore, it is unlikely that the occurrence of blowing snow differed between the site locations.

On Line 205, we added the following sentence: "The meteorological conditions for our field site, the airport site, and the ARM site are similar (generally within 0.3 m s$^{-1}$ difference for wind speed, 1 °C for air temperature, and 1% for relative humidity), as shown in Fig. S7."

[Figure]

Figure S6. The elevation map of our field site and its surroundings near Utqiaġvik, Alaska. The locations of the Field Site, the Wiley Post-Will Rogers Memorial Airport, and Department of Energy Atmospheric Radiation Measurement (ARM) Observatory are shown. This map is modified from topographyic-map.com (https://en-ng.topographic-map.com/maps/nqf6/Utqiagvik/).

[Figure]

Figure S7. Measured air temperature, relative humidity, and wind speed at the Field Site, Wiley Post-Will Rogers Memorial Airport, and Department of Energy Atmospheric Radiation Measurement (ARM) Observatory. The measurement heights at the Field Site, Airport, and ARM Observatory were 10 m, 7 m, and 8 m above the ground, respectively.

SEM-EDX analyses: The conclusion that the SSA must be emitted from bubbling bursting within open leads or from blowing snow sublimation from snow primarily impacted by sea spray aerosol deposition is based on the SEM-EDX analysis of individual particles. While this approach is appropriate, the analysis was only carried out for two days during the field work. In general, the discussion of the calcium and magnesium enrichment in sea spray aerosol arrives at conclusions that are too strong given the limited number of measurements. For example, only 7 measurements of Ca/Na are reported (Table 3), of which 3 samples have a standard deviation that is equal to the average measurement, which makes drawing conclusions difficult. After removing these weaker data points only 4 measurements remain (collected over 2 days), 3 of which exhibit significant enrichment in Ca/Na. Any conclusion regarding Ca enrichment is thus very weak and should be acknowledge as such in the text.

We acknowledge the limited number of case days for CCSEM-EDX, which is due to the labor intensive nature of these measurements and analyses. However, there is a misunderstanding in the number of data points. Since we measured the composition of individual SSA particles, there were actually 3840 measurements that each provide a Ca/Na ratio, rather than 7 measurements. This is what makes CCSEM-EDX a powerful tool for single particle analysis. In addition, the calcium enrichment was found for both the submicron and supermicron particle bulk IC analysis as well, and thus the conclusion was reached based on both bulk particle and single particle analyses.

Changes to the manuscript:
- To clarify, we added the number of SSA particles analyzed per stage in Table 3.
- On Line 279, we added the following: "We acknowledge that only four representative time periods were selected for single-particle CCSEM-EDX measurements due to the labor intensive nature of this analysis, and therefore the single particle analysis was performed in combination with bulk particle ion composition analysis in this study."
- On Line 969, we added the following: "…Both bulk and single-particle analyses show calcium enrichment relative to seawater, during moderate and observed BLSN periods…"
- On Lines 525, 526, 643, and 667, the numbers of particles analyzed by CCSEM-EDX for each of these samples are stated explicitly.

Further to this point, it would be beneficial to the manuscript to discuss in greater detail the methods used in the quantification of the elemental concentrations by SEM-EDX, especially because the ratios of elements are being considered across multiple studies. Does one need to account for differences in sensitivity to different elements? Were standards evaluated to verify the quantification?

We appreciate the question about EDX quantification and sensitivity. Converting EDX peaks to atomic % (i.e. mole %) is a standard procedure done in EDX software standard procedures that have been used for decades for a variety of applications and materials. This software accounts for sensitivity when converting from peak area to percentage, while also factoring other factors such as the Bremsstrahlung background. Sensitivity does change as a function of the energy of the X-ray, but this is accounted for using well-established conversion factors. The physics is well understood regarding the energy of these X-rays that are emitted when the electron beam knocks out the core electron, a high energy electron fills that hole in the core, and in the process to conserve energy emits a photon, which is why this can be reliably quantified. Since we know the elements in the substrates we use, these act as an internal standard (i.e. Cu in TEM grids). However, while we do cross-check this, these instruments are very stable, and the energies do not drift. We have added more detail to the description and citations to the following book chapters, initial papers using EDX quantitatively for aerosol analysis, and some of our work on this topic to provide greater context for the reader on Lines 258-262 (Goldstein et al., 2012; Fletcher et al., 2011; Laskin et al., 2002; Laskin and Cowin, 2001; Ault et al., 2012; Ault and Axson, 2017).

References:

Ault, A. P. and Axson, J. L.: Atmospheric Aerosol Chemistry: Spectroscopic and Microscopic Advances, Anal. Chem., 89, 430–452, 2017.

Ault, A. P., Peters, T. M., Sawvel, E. J., Casuccio, G. S., Willis, R. D., Norris, G. A., and Grassian, V. H.: Single-particle SEM-EDX analysis of iron-containing coarse particulate matter in an urban environment: sources and distribution of iron within Cleveland, Ohio, Environ. Sci. Technol., 46, 4331–4339, doi:10.1021/es204006k, 2012.

Fletcher, R. A., Ritchie, N. W. M., Anderson, I. M., and Small, J. A.: Microscopy and microanalysis of individual collected particles, in: Aerosol Measurement, edited by: Kulkarni, P., Baron, P. A., and Willeke, K., Wiley, printed in the USA, 179–232, 2011.

Goldstein, J., Newbury, D. E., Echlin, P., Joy, D. C., Romig Jr, A. D., Lyman, C. E., Fiori, C., Lifshin, E.: Scanning electron microscopy and X-ray microanalysis: a text for biologists, materials scientists, and geologists, Springer Science & Business Media, 2012.

Laskin, A. and Cowin, J. P.: Automated single particle SEM/EDX analysis of submicrometer particles down to 0.1µm, Anal. Chem., 73, 1023–1029, 2001.

Laskin, A., Iedema, M. J., and Cowin, J. P.: Quantitative timeresolved monitoring of nitrate formation in sea salt particles using a CCSEM/EDX single particle analysis, Environ. Sci. Tech., 36, 4948–4955, 2002.

Minor corrections and comments.

Lines 156 – 158: Is it possible that there is long-range transported anthropogenic pollution that is not correlated with $NO_X$?

Yes, it is possible. However, the pollution contribution would be at background levels. As described in Section 3.3.1, "...the atmospheric particle abundance in Utqiaġvik during low wind periods is expected to be most affected by long-range transport of aerosols contributing to springtime Arctic haze...".

Lines 300 – 302: Similar to the comment above, an alternative explanation for the apparent discrepancy is that the observations of blowing snow and the measurements of wind speed were taken at different locations.

Please refer to our first response above. We added a topography map in the supplement (Fig. S6) showing the elevation of our field site, the airport, and the ARM site in Utqiaġvik, and compared the meteorological conditions, as shown in Figure S7, for further justification of the results.

Line 369: The last sentence here is confusing, since nowhere in the preceding paragraph are the total particle number concentrations between 0.01 – 0.3 um discussed. I would recommend including a table with this data, including some of the different size fractions discussed in the text.

We added Table S5 in the supplement to show the average particle number concentrations for different size bins and wind categories, to support this discussion.

Table S5. The average (±1σ) particle number concentrations (#/cm$^3$) for different size bins and wind categories observed during the study.

| Wind category
Particle diameter | | Low wind | Moderate wind | Falsely predicted BLSN | Observed BLSN |
|---|---|---|---|---|---|
| 0.01-0.3 μm $d_m$ | 0.01-0.06 μm $d_m$ | 13±23 | 12±12 | 36±17 | 56±27 |
| | 0.06-0.3 μm $d_m$ | 131±88 | 161±63 | 193±48 | 206±37 |
| 0.3-10 μm $d_o$ | 0.3-1 μm $d_o$ | 8±7 | 12±8 | 14±4 | 12±2 |
| | 1-4 μm $d_o$ | $(2\pm3)\times10^{-1}$ | $(3\pm3)\times10^{-1}$ | $(14\pm7)\times10^{-2}$ | $(7\pm3)\times10^{-2}$ |
| | 4-5 μm $d_o$ | $(3\pm3)\times10^{-4}$ | $(2\pm2)\times10^{-4}$ | $(1\pm1)\times10^{-4}$ | $(2\pm2)\times10^{-4}$ |
| | 5-10 μm $d_o$ | $(1\pm1)\times10^{-4}$ | $(1\pm1)\times10^{-4}$ | $(1\pm1)\times10^{-4}$ | $(2\pm3)\times10^{-4}$ |

Lines 396 – 407: It would help to clarify the discussion if the authors provided some example calculations of the diameter of the blowing snow particles under the assumption of multiple SSA particles being produced from the sublimation of one larger blowing snow particle.

The calculation of the diameter of the dry SSA particles ($d_{dry}$) from blowing snow sublimation is shown in E2 on Page 4. On Lines 412-414, we show an example of calculation of $d_{dry}$ when one blowing snow particle sublimates to produce one SSA. In the case of multiple SSA particles ($N_p$) produced from the sublimation of one large blowing snow particle, $d_{dry}$ decreases to $(1/N_p)^{1/3}d_{dry}$, as shown in E2.

Following the reviewer's suggestion, we added the following on Line 412: "For example, assuming five SSA are produced from one blowing snow particle (Huang and Jaeglé, 2017; Huang et al., 2018) with salinity of 0.01 g kg$^{-1}$, blowing snow particles would have the diameters of 1-32 μm to produce SSA particles with diameters of 0.01-0.3 μm.".

Lines 438 – 439: Do the authors have any explanation why there is no observed difference between the different periods for the particles with diameters between 4-5 um? According to the information given in the text, these particles should also be scavenged by blowing snow.

This sentence was removed from the manuscript, and the sentence addressing the larger particles in the Figure 3 caption was revised to be: "Particles larger than ~4 μm were at very low concentrations and those larger than 5.3 μm experienced more than 25% sampling line losses, such that particles > 4 μm thus were not used in the discussed data analysis because of the impact of reduced statistics."

Figure 6: It is important it indicate in this figure the number of particles characterized for each size bin (n). The value of n could be indicated along the top axis of the figure. Based on the data shown in Figure S3, it seems like in the largest size bins only a few tens of particle were analyzed, but it is hard for the reader to determine that because one needs to check both the SI and Figure 6, compare size ranges, dates, etc. This is an important point because one should be cautious about drawing strong conclusions from limited measurements, if that is the case here.

Following the reviewer's suggestion, we added the total number of particles for each size bin analyzed by CCSEM-EDX to Figure 6, as shown below. Only bins with >10 particles are included in the plot.

[Figure]

Figure 6. Number fractions of seven individual particle types identified by CCSEM-EDX analysis of stage 3 ($d_a$ 0.07-0.4 μm) samples collected during (a) two moderate wind periods (April 15 19:00 – April 16 07:00 AKDT and April 16 08:00–20:00), and (b) two observed blowing snow periods (April 6 07:00–19:00 and April 6 19:00 – April 7 07:00). Total numbers of particles analyzed by CCSEM-EDX per size bin are also shown. SSA refers to sea spray aerosol. All particles were collected on the 0.07-0.4 μm $d_a$ impactor stage; projected area diameters, measured by SEM are shown. Size-resolved counts of particles analyzed by CCSEM-EDX per sample are shown in Figure S3, and representative SEM images and EDX spectra of the particle types are provided in Figure S4.

Lines 649 – 675: This section is quite long, and I think it can be trimmed to bring forward the important results suitable for a journal like ACP. The detailed comparison in the paragraph on line 649 – 666 is unnecessary as much of the information given is present in Figures 4 and 5. Also, since only 4 supermicron samples were taken I don't think the authors can draw significant conclusions by comparing the samples. The following paragraph should also be cut or reduced. It seems that the main conclusion is that bulk elemental ratios aren't representative of individual particles from different sources, which seems rather trivial.

Following the reviewer's suggestion, we removed these two paragraphs from the main text and moved some of the text to SI.

Lines 690 – 692: This sentence is confusing. What two things are similar? Please rewrite.

Open leads can produce SSA during both BLSN periods and moderate wind periods. We rewrote the sentence as: "As discussed in detail below, both submicron and supermicron inorganic SSA composition is consistent with sea spray aerosol production from open leads but with a higher flux at elevated wind speeds (Nilsson et al., 2001)".

Lines 748 – 751: I wonder how statistically significant the differences are between the sulfate/sodium ratio for the snowpack and seawater (0.04 +/- 0.01 vs. 0.06). Is this significantly lower?

From the one-sample t-test, it is statistically significant, with p value = $3\times10^{-41}$. On Line 859, we added the following: "…but statistically significantly ($p < 0.01$) lower than that of seawater (0.06; Millero et al., 2008) (Fig. 5)."

Specific comments on SI:
Formatting: Equations should be placed on their own lines and not imbedded in the text. (I am making this comment because the SI is not formatted by the journal during publication.) Also, some of the figure captions run over to the next page. If the figures and their captions were limited to one page that would make the SI easier to read.

These are now updated accordingly.

Table S1: How are the LODs determined?

The LODs for the IC analysis are based on the method calibration curve. Sample values [in µg] less than the average blank value [in µg] $+2\ \sigma$ are excluded/marked as below LOD. After correcting for the blank value, the mass measured is divided by the air volume sampled. If any of these falls below the 0.0002 $ug/m^3$ ($Na^+$, $Cl^-$) or 0.0001 $ug/m^3$ (all others), they are excluded.
Please refer to: https://saga.pmel.noaa.gov/data/stations/PMEL_Chemistry_Station_readme%20061009.txt

This information is now added to the SI.

Figure S4: Is it really possible to separate aged SSA and organic aerosol using SEM-EDX in this study? The two EDX spectra are very similar, and both C and O cannot be measured due to interferences. Also, one would expect for aged SSA and organic aerosol to be present in internally mixed particles to some extent.

It is important to define what we mean by organic aerosol. The key is that the organic aerosol category here does not have any sodium or magnesium present, which aerosol emitted from ocean water (SSA) will have (Ault et al., 2013). In Figure S4 we referred to organic aerosol as not being internally mixed with Na, Mg, K, and Ca as measured by EDX (Ault et al., 2013). It is thus straightforward to separate aged SSA (containing Na) and organic aerosol (not containing Na) from secondary or transported primary sources, as we have done in prior studies (Bondy et al., 2017; Gunsch et al., 2017; Kirpes et al. 2018). While there are some contributions to the C from the substrate, the high C peak for the organic aerosol is indicative of a particle that is primarily carbon. We have quantified C in SSA in other papers where samples were collected on silicon wafers (Kirpes et al., 2019). We now also more clearly define which type of organic aerosol we are probing in the SI (Text S2), along with further discussion and citations to clarify these points on Line 532.

Reference:

Ault, A. P., Moffet, R. C., Baltrusaitis, J., Collins, D. B., Ruppel, M. J., Cuadra-Rodriguez, L. A., Zhao, D. F., Guasco, T. L., Ebben, C. J., Geiger, F. M., Bertram, T. H., Prather, K. A., and Grassian, V. H.: Size-Dependent Changes in Sea Spray Aerosol Composition and Properties with Different Seawater Conditions, Environ. Sci. Technol., 47, 5603–5612, doi:10.1021/Es400416g, 2013.
Bondy, A. L., Wang, B., Laskin, A., Craig, R. L., Nhliziyo, M. V., Bertman, S. B., Pratt, K. A., Shepson, P. B., and Ault, A. P.: Inland Sea Spray Aerosol Transport and Incomplete Chloride Depletion: Varying Degrees of Reactive Processing Observed during SOAS, Environ. Sci. Technol., 51, 9533–9542, doi:10.1021/acs.est.7b02085, 2017.
Gunsch, M., Kirpes, R., Kolesar, K., Barrett, T., China, S., Sheesley, R., Laskin, A., Wiedensohler, A.,

Tuch, T., Pratt, K.: Contributions of Transported Prudhoe Bay Oilfield Emissions to the Aerosol Population in Utqiaġvik, Alaska, Atmos. Chem. Phys., 17, 1-29, doi:10.5194/acp-17-10879-2017, 2017.

Kirpes, R. M., Bondy, A. L., Bonanno, D., Moffet, R. C., Wang, B., Laskin, A., Ault, A. P., and Pratt, K. A.: Secondary sulfate is internally mixed with sea spray aerosol and organic aerosol in the winter Arctic, Atmos. Chem. Phys., 18, 3937-3949, doi:10.5194/acp-18-3937-2018, 2018.

Kirpes, R. M., Bonanno, D., May, N. W., Fraund, M., Barget, A. J., Moffet, R. C., Ault, A. P., and Pratt, K. A.: Wintertime Arctic sea spray aerosol composition controlled by sea ice lead microbiology, ACS Cent. Sci., 5(11), 1760-1767, doi:10.1021/acscentsci.9b00541, 2019.

Pilson, M. E. Q.: An Introduction to the Chemistry of the Sea, Cambridge University Press, 533 pp., 2013.

**Reviewer # 2**

This manuscript provides a valuable observational study that sheds light on the role of blowing snow, from sea-ice or terrestrial surfaces, for primary aerosol production in the Arctic. The authors use a combination of blowing snow observations, and bulk aerosol and single particle composition measured over an approximately 1 month period in April-May 2016. The authors conclude that if blowing snow does produce primary aerosol during observed blowing snow events, then this aerosol material must have first come from deposition of sea-spray aerosol onto snow surfaces, rather than upward migration of brine into the snowpack. This is a valuable result; however, the direct support for this conclusion from observational data is tenuous. While I recognize the significant challenge associated with collecting this dataset, the conclusions drawn may be too strong given weaknesses in the underlying data. I believe the authors can address this by more clearly articulating the potentially significant limitations in their dataset and tempering their conclusions accordingly.

Major comments:

(1) Differences in Mg/Na and Ca/Na ratios are a key result used to draw conclusions about whether blowing snow or direct SSA produced in leads is driving the enhanced aerosol numbers at high wind speeds and during blowing snow events. This data arises from both bulk aerosol composition and single particle analysis, and appropriate discussion is given to the mis-match between bulk and single particle observations. However, these data sets have some significant issues that need to be more clearly addressed in the manuscript. First, the number of individual particles corresponding to the measurements in Table 3 (and Figure 6) should be made very clear.

Following the reviewer's suggestion, we added the number of particles analyzed by CCSEM-EDX in Table 3 and Figure 6.

Second, given the large range on Mg/Na and Ca/Na ratios in Table 3 (some as large as the mean), this small number of samples cannot be used to directly conclude that aerosol during observed blowing snow events is more enriched in Calcium and thus any observed aerosol during BLSN events must have arisen from SSA deposition rather than brine migration.

Although only 7 sample substrates were used for CCSEM-EDX analysis, there were actually 3840 single particle measurements, since we measured the composition of individual SSA particles. In addition to calcium enrichment observed in these individual SSA measurements, the calcium enrichment was also found for the bulk submicron and supermicron particle samples as well. Thus, the conclusion was reached based on both bulk particle and single particle analyses. On Lines 47, 884, and 991, we changed the phrasing "must have been" to "would have been".

Third, in Figure 5, are the differences between Ca/Na ratios in snow-pack between the different BLSN and wind-speed periods statistically significant? The 1-sigma range of measurements are large, while the absolute differences are fairly small for both sub-micron aerosol and tundra snowpacks. For these three reasons, the conclusions drawn are tenuous at best, and should be described as such in the manuscript.

In Figure 5, the tundra surface snowpack $[Ca^{2+}]/[Na^+]$ molar ratios are 0.033±0.004, 0.025±0.007, 0.2±0.2, and 0.2±0.2 during observed BLSN, falsely predicted BLSN, moderate wind, and low wind periods (Table S3). Statistical t-tests were performed for each two within these four categories, and all show significant differences at 95% confidence levels, except for the comparison between moderate wind and low wind periods. Thus, the tundra surface snowpack $[Ca^{2+}]/[Na^+]$ molar ratio during both observed BLSN and falsely predicted BLSN is significantly different from the moderate wind and low wind periods, as shown on Line 835.

(2) From previous observations of blowing snow, what is the approximate horizontal extent? Is it possible that this is heterogeneous on the scale of 5-10's of km? From comparison of airport and ceiliometer BLSN observations, this does appear to be the case. The implications of this for diagnosing blowing snow in source regions to the author's sampling site should be clearly described.

The horizontal extent of blowing snow is generally more than a few kilometers (Tabler et al., 1990). To address comparison of the airport and ARM observations, we added a topography map in the supplement (Fig. S6) showing the elevation of our field site, the airport, and the ARM site in Utqiaġvik. The elevation difference between these three sites is within 5 m. In addition, the surface type within the area of these three sites is mainly flat tundra. The difference in the measured meteorological parameters was generally within 0.3 m s$^{-1}$ for wind speed, 1 °C for air temperature, and 1% for relative humidity between these three sites, as now shown in Figure S7. Therefore, it is unlikely that the spatial difference of the sites caused the blowing snow to be falsely predicted. On Line 205, we added the following sentence: "The observed meteorological conditions for our field site, the airport site, and the ARM site were similar (within 0.3 m s$^{-1}$ difference for wind speed, 1 °C for air temperature, and 1% for relative humidity), as shown in Fig. S7.". We added the new Figures S6 and S7 to the SI. For more details, please refer to our first response to the first reviewer.

Reference:
Tabler, R. D., Pomeroy, J. W., and Santana, B. W.: Drifting snow, in: Cold Regions Hydrology and Hydraulics, Technical Council on Cold Regions Engineering Monograph, edited by: Ryan, W. L., Crissman, R.D., American Society of Civil Engineers, New York, 95-145, 1990.

(3) The manuscript is generally well written; however, the Results & Discussion section corresponds to a very detailed description of the results only, with very little interpretation or guidance given to the reader. In conjunction with this, the Conclusions section is very long, and much of the interpretation that would guide the reader through the at times complex results, resides in the Conclusions. The manuscript will benefit from (1) restructuring the Results & Discussion and Conclusions section to better describe the main outcomes and interpretation of the dataset, and (2) focused discussion of the limitations, potentially within its own section of the Results & Discussion.

Following the reviewer's suggestion, we revised these two sections and made the Results & Discussion more concise by moving some of the text to SI. In addition, as noted in previous comment responses, we added statements of statistical significance tests, as well as reference to the hundreds of individual particles analyzed per substrate by CCSEM-EDX, to strengthen and more clearly communicate the results.

Specific comments.
L423-426: This suggests lower concentrations between 1-2 um (not 1-4um), and the conclusion that these particles are "removed" by blowing snow is based on observation of a correlation, rather than direct causation. The calculations described in the following sentences do support this conclusions, but the authors should be clear about the limitations in their ability to assert a mechanism.

From the OPS measurements of larger particles, for each size bin ranging from 1 to 4 μm $d_o$ (6 size bins), the average particle number concentrations during observed BLSN periods were significantly ($p < 0.05$) lower than during moderate wind conditions and low wind periods ($p < 0.05$). Compared to falsely predicted BLSN periods, the average particle number concentrations during observed BLSN periods were significantly lower ($p < 0.05$) for 1-2 μm $d_o$ (3 size bins), but not significantly different ($p > 0.05$) for 2-4 μm $d_o$ (3 size bins). We appreciate the reviewers point about correlations versus causation. Therefore, on Line 442, we replaced the sentence "…This suggests a net removal of 1-4 μm do particles by blowing snow…" with "…This suggests that supermicron particles may be removed by BLSN…".

Figure 4: Given the variability in many of these observations, rather than showing only the average +/- 1 standard deviation of these measurements, it may be more useful to the reader to show e.g., a box and whisker plot, with all data points (or violin plot) shown behind.

We appreciate the suggestion. However, after considering this option, we found that we prefer the simplified version of Figure 4 to better communicate the trend and result.

Section 3.3.2: A large amount of space is dedicated here to establishing the importance of Arctic Haze during moderate wind periods. This is well known, may not be supporting the main conclusions of the manuscript. I suggest the authors consider whether this detailed description is needed in the main text.

Following the reviewer's suggestion, we moved some of this text to SI.

Figure 6: The total number of sampled particles in each size bin should be overlaid on top of the composition data in both panels of this figure.

Following the reviewer's suggestion, we added the total number of particles in each size bin analyzed by CCSEM-EDX in Figure 6.

Section 3.3.3: Please be more clear about where the absolute number of individual particles collected are presented within this section (presumably this corresponds to data presented in Figure S3?). This discussion is challenging to follow without a summary of these numbers and clear indication of how many individual particles these percentages are based upon. Further, a large amount of space is given here (and in lines 736-746, Section 3.3.4) to establishing that bulk aerosol composition is not representative of individual aerosol mixing state. Since this is relatively well understood in general, I questions whether such a large amount of text is required to describe this. It is certainly important, and can likely be gleaned from a short discussion and reference to the bulk and single particle data figures.

The total number of particles in each size bin analyzed by CCSEM-EDX is now included in Figure 6. The number of SSA analyzed by CCSEM-EDX per sample is now included in Table 3.

As discussed in the response to the first reviewer, we shortened Section 3.3.3, and moved some of the text to SI. Following the reviewer's suggestion, Lines 736-746 specifically were moved to SI (S1.2).

L692-694: It is unclear how this final statement follows from the information covered in this paragraph. Please revise.

The sentence was deleted.

Lines 748-752: Are these differences in SO4/Na ratios statistically significant?

From the one-sample t-test, the average $[SO_4^{2-}]/[Na^+]$ molar ratio of the local tundra surface snowpack during observed BLSN periods (0.04±0.01) is statistically significantly lower than that of seawater (0.06), with a p value = $3 \times 10^{-41}$. On Line 859, we added the following: "…statistically significantly ($p < 0.01$) lower than that of seawater (0.06; Millero et al., 2008) (Fig. 5)...". The other two $[SO_4^{2-}]/[Na^+]$ ratios (0.04 ± 0.01 and 0.03 ± 0.01 for 0.4-2.8 μm $d_a$ SSA on April 6 and April 15-16) are not statistically significantly different.

Lines 833-835 (and in general): Has complete lack of Ca enrichment during upward brine migration been shown in previous literature, or do the authors assume that no enrichment occurs in the brine migration process?

We don't expect calcium enrichment in the brine migration process, as discussed by Hara et al. (2017). We now cite this publication on Line 975.

Reference:
Hara, K., Matoba, S., Hirabayashi, M., and Yamasaki, T.: Frost flowers and sea-salt aerosols over seasonal sea-ice areas in northwestern Greenland during winter–spring, Atmos. Chem. Phys., 17, 8577–8598, doi:10.5194/acp-17-8577-2017, 2017.